# LLMs Get Lost In Multi-Turn Conversation

**Philippe Laban**[*◇]     **Hiroaki Hayashi**[*♣]     **Yingbo Zhou**[♣]     **Jennifer Neville**[◇]

◇Microsoft Research     ♣Salesforce Research

{plaban,jenneville}@microsoft.com
{hiroakihayashi,yingbo.zhou}@salesforce.com

## ABSTRACT

Large Language Models (LLMs) are conversational interfaces. As such, LLMs have the potential to assist their users not only when they can fully specify the task at hand, but also to help them define, explore, and refine what they need through multi-turn conversational exchange. Although analysis of LLM conversation logs has confirmed that underspecification occurs frequently in user instructions, LLM evaluation has predominantly focused on the single-turn, fully-specified instruction setting. In this work, we perform large-scale simulation experiments to compare LLM performance in single- and multi-turn settings. Our experiments confirm that all the top open- and closed-weight LLMs we test exhibit significantly lower performance in multi-turn conversations than single-turn, with an average drop of 39% across six generation tasks. Analysis of 200,000+ simulated conversations decomposes the performance degradation into two components: a minor loss in aptitude and a significant increase in unreliability. We find that LLMs often make assumptions in early turns and prematurely attempt to generate final solutions, on which they overly rely. In simpler terms, we discover that **when LLMs take a wrong turn in a conversation, they get lost and do not recover**.

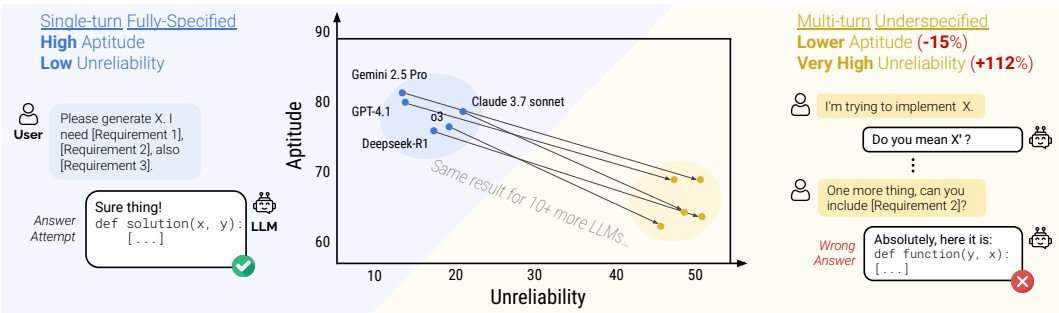

Figure 1: Our simulated conversations for 6 generation tasks on the 15 LLMs observe a major performance drop in multi-turn settings (-39%), explained by some loss in Aptitude, and large loss in Reliability.

## 1 INTRODUCTION

Today's large language models (LLMs) function as conversational interfaces (*e.g.*, ChatGPT, Gemini, Claude), enabling users to interact with the LLM through multiple conversation turns. Such interaction promises to help users not only when they know what they need (i.e., they can fully specify their requirements in an instruction), but also when they don't. In such cases, users might start with an underspecified instruction and further clarify their needs through turn interactions. Though studies of LLM conversation logs have confirmed that underspecification in user instructions is prevalent (Herlihy et al., 2024), LLM systems are typically evaluated in single-turn, fully-specified settings.

Even though a growing body of work proposes to evaluate LLMs in a **multi-turn** fashion, we identify in our review (Section 2) that most prior work treats the conversation as *episodic*: conversation turns

---

[*]Equal Contributions

might relate to each other, but the conversation can effectively be decomposed as an array of subtasks that can be evaluated in isolation. We argue that episodic tasks move away from what is prevalent in human conversation: underspecification (Zipf, 1949; Herlihy et al., 2024).

In this work, we close this gap by creating a simulation environment for multi-turn underspecified conversations – sharded simulation – that leverages existing instructions from high-quality single-turn benchmarks. At a high level, the sharding process we propose transforms existing single-turn instructions into *sharded instructions*, a set of smaller instructions that jointly deliver the same information as the original instruction. Sharded simulation then ensures that each turn of conversation reveals at most one shard of information per conversation turn, enforcing that the instruction is gradually revealed through the conversation.

On the set of tasks that we experimented on, we observed that models engaged in multi-turn underspecified conversations achieved an average performance of 65%–a 25-point drop from single-turn performances of 90% when they receive the entire instruction at the beginning of the conversation. Notably, we observe this drop in performance even in two-turn conversations, and across all LLMs we test, from small open-weights (LLama3.1-8B-Instruct) to state-of-the-art (Gemini 2.5 Pro).

Furthermore, we decompose the performance degradation into two components: (1) loss in aptitude, and (2) increase in unreliability. We find that in single-turn, LLMs with higher aptitude tend to be more reliable (*e.g.*, GPT-4.1). On the other hand, all LLMs exhibit very high unreliability in multi-turn settings, regardless of aptitude. We refer to this as the *lost in conversation phenomenon*: when LLMs take a wrong turn in multi-turn conversation, they get lost and do not recover.

We investigate several explanations for this effect and show that the LLMs tend to (1) generate overly verbose responses, leading them to (2) propose final solutions prematurely in conversation, (3) make incorrect assumptions about underspecified details, and (4) rely too heavily on previous (incorrect) answer attempts.

Our findings highlight a gap between how LLMs are used in practice and how the models are being evaluated. Ubiquitous performance degradation over multi-turn interactions is likely a reason for low uptake of AI systems (Southworth et al., 2023; Brauner et al., 2023; Horowitz et al., 2024), particularly with novice users who are less skilled at providing complete, detailed instructions from the onset of conversation (Zamfirescu-Pereira et al., 2023; Knoth et al., 2024). We provide actionable recommendations based on small-scale experiments and make a concrete call-to-action to LLM builders, urging them to prioritize multi-turn reliability in conjunction with aptitude.

## 2 BACKGROUND AND RELATED WORK

Previous-generation language models (e.g., BART (Lewis et al., 2019), GPT-2 (Radford et al., 2019), or T5 (Raffel et al., 2020)) were designed for single-turn tasks, and so do the evaluation. Conversational agents were instead built as modular systems using language models as a component (Konrád et al., 2021), and were evaluated through human protocols (Deriu et al. 2021; Murakhovs' ka et al. 2023, *inter alia*).

While a series of work on multi-turn evaluation emerged since the rise of ChatGPT (Zheng et al., 2023b; Kwan et al., 2024), we argue that such works frame conversation as *episodic*: every turn is a self-contained subtask that can be graded in isolation. We show that such a design overestimates LLM capability. In fact, when the model must fuse dispersed clues across turns, performance drops sharply and consistently (Appendix G.3). Indeed, real users often issue **underspecified** instructions both in human-AI communication (Herlihy et al., 2024) and in natural human communication–named as "principle of least effort" (Zipf, 1949). We place underspecification at the center of our study.[1]

A second limitation of prior work on multi-turn episodic evaluation is the mismatch in task granularity: multi-turn subtasks differ from the single-turn counterpart, which prevents direct comparison. We run both settings on *a common set of tasks*, allowing a precise observation of performance degradation from single to multi-turn. Crucially, we focus on open-ended generation–the predominant real-world use case in code and natural-language tasks (Zheng et al., 2023a; Handa et al., 2025)–rather than short-form generation or classification.

---

[1]Appendix A reviews related work specifically focused on underspecified communication.

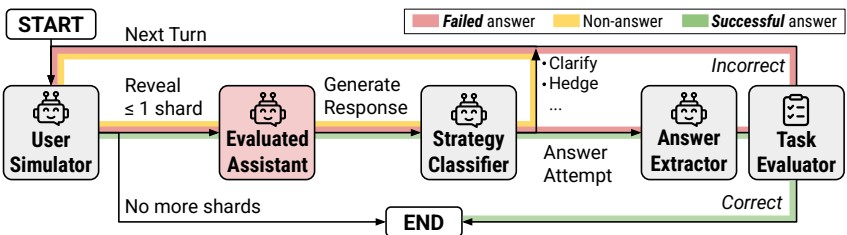

Figure 2: Sharded Conversation Simulation Diagram. The simulation subject is highlighted in red.

Scaling multi-turn experimentation requires simulating a user. Possible options range from templates (Choi et al., 2018; Reddy et al., 2019; Laban et al., 2023; Deng et al., 2024), LLM (Poelitz & McKenna, 2025; Li et al., 2024; Chang et al., 2025; Liang et al., 2024), fixed annotations (Finch et al., 2023; Chang et al., 2025), or real users (Ram et al., 2018; Laban et al., 2021; Chiang et al., 2024), each trading realism for cost and scalability. We adopt an LLM-based simulator that balances diversity and control, treating it strictly as a probe of *LLM behavior*, not user behavior (due to the approximation on the user end). In addition, our simulation is simplistic and idealized, and the degradations we report likely underestimate those in genuine human-AI conversations (Section 8).

## 3 SIMULATING UNDERSPECIFIED, MULTI-TURN CONVERSATION

We develop a simulation environment that repurposes existing tasks from single-turn benchmarks. First, we apply a *sharding process* to transform original fully-specified instructions into *sharded instructions*. Second, we implement a sharding simulation environment that carries out a multi-turn conversation based on a sharded instruction.

### 3.1 SHARDING PROCESS

To construct high-quality sharded instructions from the original fully-specified instructions, we define a set of properties that a sharded instruction must satisfy, such as information preservation, clear initial intent, and order insensitivity. We refer to Appendix B for a precise and mathematical definition of shards and their properties. Based on these properties, we develop a semi-automatic sharding process to scale the creation of sharded instructions, which involves (1) segmentation (2) rephrasing (3) verification (4) manual inspection.[2] At a high level, a sharded instruction is composed of *a set of shards*, each introducing a single element from the original instruction. Taken jointly, the set of shards reflects the same information provided in the fully-specified instruction, with the information explicitly divided across shards. During manual inspection, we reviewed every sharded instruction, merging, splitting, or reordering shards to enhance the naturalness of the sharded sample, and editing phrasing to ensure that each shard represents a natural unit of information a user might reasonably provide in a single turn and not an adversarial representation of the original instruction. This process ensures that the experiments we carried out used sharded instructions that adhered to the properties we defined. Example pairs of fully-specified and sharded instructions are shown in Figure 4.

### 3.2 SIMULATING SHARDED CONVERSATIONS

Figure 2 outlines our multi-turn simulator for sharded tasks. We implement a loop with three LLM-backed roles. The **assistant** is the model under test, the **user** owns the full sharded instruction and chooses which shard to disclose each turn, and the **system** tags and grades assistant replies.

At turn 1 the user reveals Shard 1, the assistant responds freely, and the system maps that reply to one of seven strategies—*clarification*, *refusal*, *hedging*, *interrogation*, *discussion*, *missing*, or *answer attempt*—following Herlihy et al. (2024). If a reply contains an answer attempt, we extract the answer span (e.g., a number or code block) to shield scoring from surrounding text, then pass it to a task-specific evaluator. Subsequent turns repeat: the user may expose one additional shard, the assistant replies, and any answer attempt is scored. A conversation's final score is the maximum over all per-turn scores: in a conversation with $N$ shards, the assistant gets up to $N$ answer attempts, and

---

[2]The sharding process is described in depth in Appendix C.

is credited for the best one. From that perspective, the assistant has some advantage in the sharded setting over single-turn simulations as they only permit at most one answer attempt. The conversation ends when an answer is deemed correct or no shards remain.

Choosing the next shard is non-trivial because assistants often ask shard-specific follow-ups. We therefore instantiate the user simulator with an LLM (GPT-4o-mini), giving it the entire instruction and conversation history so it can select and lightly rephrase the shard that best fits the exchange, for instance, responding to a clarification question with the relevant shard rather than revealing shards in a fixed or random order. Before the first turn, the assistant receives only minimal context (e.g., a tool list) as a system prompt; it is never told that the conversation will be underspecified or multi-turn, enabling us to measure default behavior.

The strategy classifier and answer extractor are also prompt-based GPT-4o-mini modules. The classifier's primary role in the simulation loop is to detect *answer attempt* turns, which trigger answer extraction and scoring; it does not directly influence the user simulator's shard selection. To ensure the validity of the framework, we manually reviewed several hundred conversations and found that errors in any component occurred in fewer than 5% of cases, and less than 2% of cases where the assistant was disfavored.[3] We thus regard the simulator as sufficiently accurate for our experiments.

Appendix K provides an example simulated sharded conversation.

## 3.3 SIMULATION TYPES

We simulate five types of conversation, each with a different pace of information:

📄 **FULLY-SPECIFIED** (short-form: FULL) Single-turn task: the original instruction appears in turn 1. This serves as the baseline.

🧩 **SHARDED** Multi-turn task: the sharded instruction is revealed across turns as described above. Core setting for underspecification.

🗄 **CONCAT** All shards are concatenated into one bullet-point prompt in a single turn. This setting removes underspecification (like FULL) while retaining sharding rephrasing; a control to ensure the performance drop in SHARDED is due to multi-turn underspecification, not due to rephrasing.

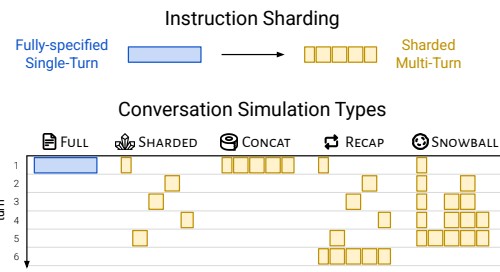

Figure 3: Conversation simulation types based on sharded instructions. Once a fully-specified instruction (blue block) is sharded (yellow blocks), the "shards" can be used to simulate single-turn (FULL, CONCAT) or multi-turn (SHARDED, RECAP, SNOWBALL) conversations, affecting the pace of information disclosure.

🔄 **RECAP** runs a SHARDED conversation then adds one recap turn that restates every shard, giving the model a last, "fully-specified" chance. This setting tests whether a simple agent-style recap mitigates SHARDED degradation.

😊 **SNOWBALL** At each turn, the user reveals the next shard and repeats all prior shards, "snowballing" the context. This setting evaluates whether continual reminders reduce the memory burden over long, multi-turn interactions.

# 4 EXPERIMENT

## 4.1 TASK SELECTION

We construct sharded instructions for six tasks that we use in a large-scale simulation experiment. For each task, we select instructions from one or two high-quality single-turn benchmarks and apply a semi-automatic sharding process (outlined in Appendix C). For each task, we prepare 90-120 sharded instructions. We select popular and diverse generation tasks across programming and non-programming use cases. Figure 4 provides an example of an original and sharded instruction for each task, which we introduce below:

---

[3]See Appendix D for details on the annotation process.

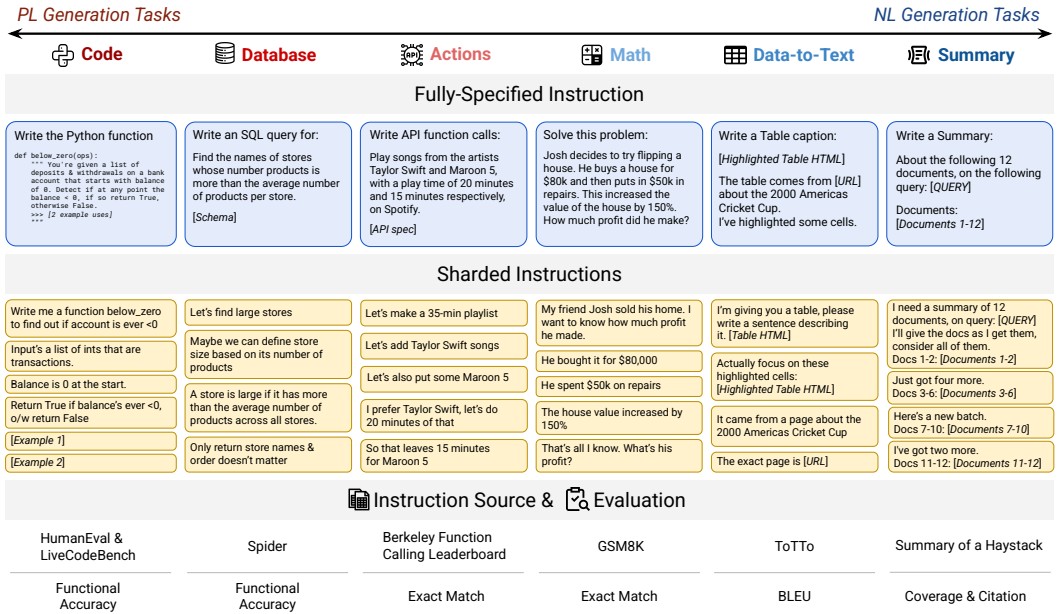

Figure 4: Six sharded tasks included in our experiments. We include tasks that involve generating programming and natural language. For each task, an illustrative fully-specified instruction and its sharded counterpart. We sharded 90 to 120 instructions based on existing datasets (Instruction Origin), and re-purposing evaluation.

**Code**   The assistant writes a Python function that implements the instruction. The original instructions are sourced from HumanEval (Chen et al., 2021) and LiveCodeBench (Jain et al., 2024).

**Database**   The assistant generates an SQL query given a database's schema and a user query in natural language (Text-to-SQL). The original instructions and databases are sourced from the popular Spider dataset (Yu et al., 2018).

**Actions**   The assistant generates API calls that satisfy a user request given a set of API schemas. We source API schemas and user instructions from Berkeley Function Calling Leaderboard (BFCL) (Yan et al., 2024), a standard benchmark for measuring function calling capabilities.

**Math**   The assistant numerically solves an elementary math word problem. Problems are sourced from the GSM8K dataset (Cobbe et al., 2021).

**Data-to-text**   The assistant produces a caption describing a table and several elements of related metadata. The ToTTo dataset (Parikh et al., 2020) is used to prepare sharded instructions.

**Summary**   The assistant generates a summary with citations given a user query and a set of (around twenty) documents. We re-purpose the instructions from Summary of a Haystack (Laban et al., 2024). The instruction in this task requires long-context understanding, a skill models are known to struggle with (Huang et al., 2023; Karpinska et al., 2024; Kim et al., 2024a).

For each task, we reuse the metrics used in the original benchmarks to assess instance-level correctness. We measure Code and Database with the functional accuracy, Actions and Math with semantic equivalence to the reference answer, all of which render a binary correctness. Data-to-Text and Summary are *refinement tasks*, which get scored on a range (0-100): BLEU (Papineni et al., 2002) for Data-to-Text and a custom LLM-as-a-judge metric ("Joint Score") for Summary. We map binary accuracy to the range of 0-100 (0 = failure, 100 = success) so that all tasks produce scores on a common scale, facilitating aggregation.[4]

---

[4]Appendix J lists task-specific sharding details

## 4.2 Simulation Metrics

LLMs employ a stochastic process to generate text. When setting the generation parameters to their default (*e.g.*, temperature = 1.0), they generate many distinct responses for a fixed conversation state. We leverage this property to conduct repeated simulations for a given instruction and quantify the variations that occur. Suppose the $i$-th simulation yields a score $S_i \in [0, 100]$ from a task-specific evaluator. By running $N$ simulations for an instruction and obtaining the set of scores $S = \{S_i\}_{i=1}^{N}$, we define three metrics: **averaged performance** $\overline{P}$, **aptitude** $A^{90}$, and **unreliability** $U_{10}^{90}$:

$$\overline{P} = \sum_{i=1}^{N} S_i / N, \quad A^{90} = \mathrm{percentile}_{90}(S), \quad U_{10}^{90} = \mathrm{percentile}_{90}(S) - \mathrm{percentile}_{10}(S).$$

Averaged performance $\overline{P}$ is an unbiased estimate of a model's mean score on an instruction in a given simulation type. Aptitude $A^{90}$ is an estimate of a model's 90th percentile score on a given instruction, *i.e.* a best-case metric that estimates scores obtained in the top 10% of simulations conducted. Unreliability $U_{10}^{90}$ is an interpercentile range estimate measuring the gap between the 90th and 10th percentile estimates, giving a sense of *level of degradation* that occurs in response quality due to stochasticity in the LLM.[5]

Each metric is computed on a per-instruction basis and can be averaged across a corpus of instructions to obtain corpus-level metrics.

For the rest of the paper, we refer to reliability and unreliability interchangeably, with reliability defined as $R_{10}^{90} = 100 - U_{10}^{90}$. We also simplify the notations to $A$ for aptitude and $U$ for unreliability, though the metrics can be generalized to other percentile thresholds (*e.g.*, $A^{80}$ or $U_5^{95}$). Figure 5a visually connects the aptitude and unreliability metrics to box-plot visualizations. The height of the upper whisker of the box plot represents aptitude (A), and the distance between the upper and lower whiskers represents Unreliability (U).

## 4.3 Simulation Scale and Parameters

In the main simulation experiment, we simulate conversations across three types: 📄 FULL, 🗄 CONCAT, and 🔱 SHARDED on around 100 instances for each of the six tasks. We experiment with 15 LLMs, running $N = 10$ simulations for each pair of model and simulation type, totaling more than 200,000 simulated conversations. All simulations are conducted with a default temperature of $T = 1$, however, we performed a supplementary experiment (Section G.2) that explores the effect of temperature on aptitude and reliability. Although simulating ten conversations for each (`LLM, instance, simulation type`) increases experimental costs ten-fold, it allows us to not only measure averaged performance ($\overline{P}$) more accurately, but also study aptitude and reliability of LLM systems in depth in Section 5.2.

We select 15 LLMs from eight model families: 🌀 OpenAI (GPT-4o-mini, GPT-4o (Hurst et al., 2024), o3 (OpenAI, 2025), and GPT-4.1), 🅰 Anthropic (Claude 3 Haiku, Claude 3.7 Sonnet), Google's ◆Gemini (2.5 Flash, 2.5 Pro; Team et al. 2023), Meta's ∞ Llama (Llama3.1-8B-Instruct, Llama3.3-70B-Instruct, Llama 4 Scout; Grattafiori et al. 2024), ✦ AI2 OLMo-2-13B (OLMo et al., 2024), ⊞ Microsoft Phi-4 (Abdin et al., 2024), 🐋 Deepseek-R1 (Guo et al., 2025), and 🔴 Cohere Command-A (Cohere et al., 2025). The selection prioritizes the evaluation of state-of-the-art models: small (8B) to large (300B+), open- to closed-weights, and two reasoning models (o3, R1) to probe test-time compute in multi-turn conversations. Details on model versioning and access are listed in Appendix I. We estimate the total cost of conducting simulations to be around $5,000.

All tasks in our experiments require fewer than 20k tokens of total context, with the summarization task requiring the most and all others typically fitting within 8k tokens. Two models (Phi-4, OLMo-2-13B) could not accommodate the summarization task's context length and were excluded from it (indicated by dashes in Table 1). We emphasize that the intent of these experiments is to study multi-turn behavior at *regular* context lengths, not to stress-test long-context capabilities.

| Model | FULL | | | | | | CONCAT | | | | | | SHARDED | | | | | | Overall | |
|---|---|---|---|---|---|---|---|---|---|---|---|---|---|---|---|---|---|---|---|---|
| | ⊕ | 🗄 | ⚙ | ▦ | 📑 | 📋 | ⊕ | 🗄 | ⚙ | ▦ | 📑 | 📋 | ⊕ | 🗄 | ⚙ | ▦ | 📑 | 📋 | 🖴/📄 | 🪷/📄 |
| 3.1-8B | 27.4 | 64.1 | 82.9 | 13.7 | 63.9 | 7.6 | 21.2 | 47.7 | 83.0 | 15.7 | 62.6 | 6.5 | 21.7 | 25.9 | 45.5 | 13.3 | 37.4 | 3.4 | 91.6 | 62.5 |
| OLMo2 | 18.8 | 54.8 | 56.1 | 17.2 | 80.0 | - | 16.3 | 40.5 | 49.8 | 14.3 | 80.1 | - | 14.4 | 22.4 | 13.8 | 9.0 | 46.3 | - | 86.5 | 50.5 |
| 3-Haiku | 44.8 | 85.0 | 83.5 | 29.8 | 73.9 | 11.6 | 36.3 | 76.5 | 80.2 | 30.1 | 76.1 | 9.2 | 31.5 | 31.8 | 55.9 | 18.6 | 47.1 | 1.6 | 91.6 | 52.4 |
| 4o-mini | 75.9 | 89.3 | 94.1 | 35.9 | 88.1 | 14.9 | 66.7 | 90.7 | 92.2 | 31.2 | 88.0 | 12.5 | 50.3 | 40.2 | 52.4 | 19.8 | 58.7 | 7.2 | 93.0 | 56.2 |
| 3.3-70B | 72.0 | 91.1 | 95.0 | 34.1 | 91.7 | 15.8 | 52.7 | 87.9 | 97.0 | 32.0 | 91.8 | 14.7 | 51.6 | 35.4 | 71.0 | 22.4 | 61.5 | 10.5 | 93.2 | 64.2 |
| Phi-4 | 53.2 | 87.6 | 82.7 | 23.9 | 89.2 | - | 48.4 | 79.6 | 76.0 | 28.6 | 90.4 | - | 39.1 | 33.1 | 34.1 | 23.2 | 52.5 | - | 99.0 | 61.7 |
| CMD-A | 72.0 | 91.9 | 98.5 | 27.7 | 94.5 | 24.3 | 61.6 | 86.1 | 98.4 | 33.2 | 91.9 | 21.3 | 44.9 | 33.6 | 72.0 | 27.9 | 66.0 | 4.9 | 97.3 | 60.4 |
| 4-Scout | 73.9 | 92.7 | 98.0 | 35.2 | 96.3 | 13.7 | 60.3 | 81.5 | 98.3 | 28.2 | 92.9 | 13.7 | 46.4 | 27.1 | 69.9 | 26.1 | 67.0 | 12.3 | 91.0 | 66.1 |
| o3 | 86.4 | 92.0 | 89.8 | 40.2 | 81.6 | 30.7 | 87.2 | 83.3 | 91.5 | 39.4 | 80.0 | 30.4 | 53.0 | 35.4 | 60.2 | 21.7 | 63.1 | 26.5 | 98.1 | 64.1 |
| 3.7-Sonnet | 78.0 | 93.9 | 95.4 | 45.6 | 85.4 | 29.3 | 76.2 | 81.5 | 96.0 | 53.3 | 87.2 | 28.9 | 65.6 | 34.9 | 33.3 | 35.1 | 70.0 | 23.6 | 100.4 | 65.9 |
| R1 | 99.4 | 92.1 | 97.0 | 27.0 | 95.5 | 26.1 | 97.1 | 89.9 | 97.0 | 36.7 | 92.9 | 24.4 | 70.9 | 31.5 | 47.5 | 20.0 | 67.3 | 17.2 | 103.6 | 60.8 |
| 4o | 88.4 | 93.6 | 96.1 | 42.1 | 93.8 | 23.9 | 82.9 | 91.7 | 97.1 | 32.2 | 91.9 | 23.9 | 61.3 | 42.3 | 65.0 | 20.5 | 67.9 | 10.6 | 94.5 | 57.9 |
| 2.5-Flash | 97.0 | 96.3 | 88.4 | 51.2 | 90.6 | 29.1 | 92.5 | 95.5 | 89.2 | 51.9 | 89.6 | 29.4 | 68.3 | 51.3 | 42.6 | 31.0 | 66.1 | 26.1 | 99.3 | 63.8 |
| 4.1 | 96.6 | 93.0 | 94.7 | 54.6 | 91.7 | 26.5 | 88.7 | 86.5 | 98.5 | 54.4 | 89.7 | 26.8 | 72.6 | 46.0 | 62.9 | 28.6 | 70.7 | 13.3 | 97.9 | 61.8 |
| 2.5-Pro | 97.4 | 97.3 | 97.8 | 54.8 | 90.2 | 31.2 | 95.7 | 94.9 | 98.1 | 56.9 | 89.3 | 31.8 | 68.1 | 43.8 | 36.3 | 46.2 | 64.3 | 24.9 | 100.1 | 64.5 |

Table 1: Averaged Performance ($\overline{P}$) of LLMs on six tasks (⊕ Code, 🗄 Database, ⚙ Actions, ▦ Data-to-text, 📑 Math, and 📋 Summary). For each task, conversations are simulated in three settings: 📄 FULL, 🖴 CONCAT, and 🪷 SHARDED. Models are sorted in ascending order of average FULL scores across tasks. Background color indicates the level of degradation from the FULL setting. The last two columns average the performance drops from the CONCAT and SHARDED compared to the FULL in percentages across the six tasks.

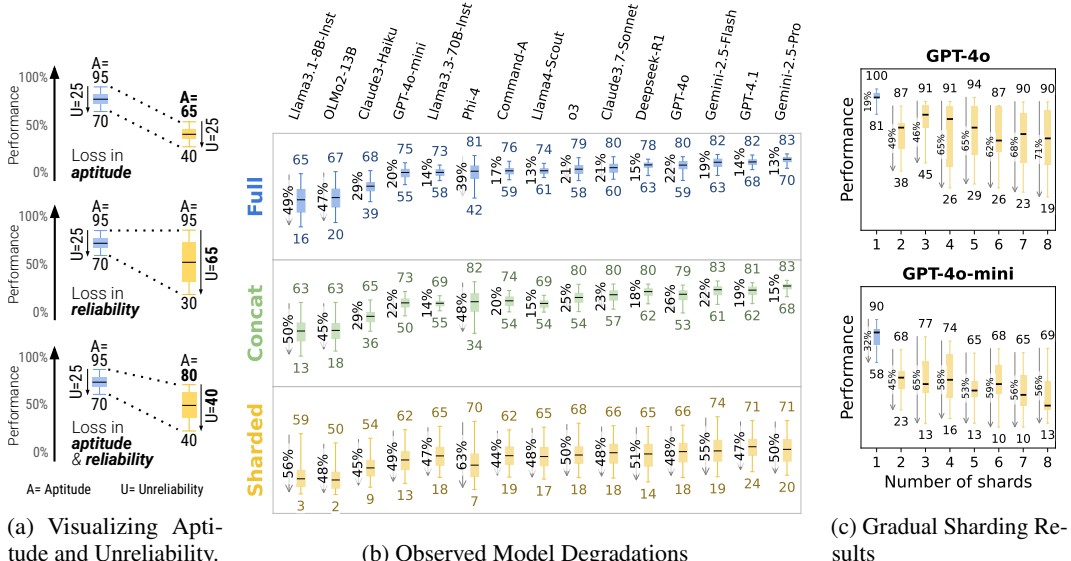

(a) Visualizing Aptitude and Unreliability.

(b) Observed Model Degradations

(c) Gradual Sharding Results

Figure 5: (a) Visual introduction to the concepts of Aptitude and Unreliability when overlaid on a box-plot visualization, (b) reliability results based on experimental simulations with 15 LLMs, (c) summary of results from gradual sharding experiment, with instructions sharded in gradually larger shard sets (from 1 to 8 shards).

# 5 RESULTS

## 5.1 AVERAGE PERFORMANCE FINDINGS

Table 1 summarizes results from the simulation. At a high level, **every model sees its performance degrade on every task when comparing FULL and SHARDED performance, with an average degradation of -39%**. We name this phenomenon Lost in Conversation: models that achieve stellar (90%+) performance in the lab-like setting of fully-specified, single-turn conversation struggle on *the exact same tasks* in more realistic, underspecified, and multi-turn conversations.

---

[5]Appendix E shows how a drop in performance from 90% to 60% can stem from aptitude loss, reliability issues, or both, depending on the error breakdown.

In comparison, models perform roughly equivalently in the CONCAT setting, with CONCAT performance averaging 95.1% of the FULL counterpart. This implies that the loss in performance for SHARDED is not explained by potential loss of information in sharded instructions, as such a loss would be reflected in lower CONCAT performance. We observe that smaller models (Llama3.1-8B-Instruct, OLMo-2-13B, Claude 3 Haiku) have more pronounced CONCAT degradations (86-92), and interpret this as indicating that smaller models struggle to generalize as well as larger models: benign rephrasing affects performance more than for larger, more robust models. This lack of robustness to paraphrasing can be observed visually in Table 1: CONCAT degradation (red background) is more pronounced in the top rows (weaker models) than the bottom rows (stronger models).

The last column of the Table (🏵 / 📄) aggregates performance degradation across the six tasks, summarizing the magnitude of the Lost in Conversation effect for each model. Surprisingly, **more performant models (Claude 3.7 Sonnet, Gemini 2.5, GPT-4.1) get equally lost in conversation compared to smaller models (Llama3.1-8B-Instruct, Phi-4)**, with average degradations of 30-40%. This is in part due to metric definitions. Since smaller models achieve lower absolute scores in FULL, they have less scope for degradation than the better models. In short, no matter how strong an LLM's single-turn performance is, we observe large performance degradations in the multi-turn setting.

When looking at the task-specific breakdown, some models see more muted degradations in certain tasks. For instance, Command-A sees the least degradation on Actions, while Claude 3.7 Sonnet and GPT-4.1 conserve performance well on Code, and Gemini 2.5 Pro in Data-to-Text. This finding indicates that the multi-turn capabilities of models are not uniform across domains and validates the importance of benchmarking models across a wide variety of tasks to investigate model capabilities.

Additional test-time compute (reasoning tokens) does not help models navigate multi-turn underspecification, as the two reasoning models (o3, Deepseek-R1) deteriorate in similar ways to non-reasoning models. This result confirms that **additional test-time compute does not, on its own, allow models to strategize over multi-turn conversation**. The analysis we conduct identifies a potential root cause: reasoning models tend to generate lengthier responses (on avg. 33% longer than non-reasoning LLMs). As we find in Appendix F, longer assistant responses tend to contain more assumptions, which can confuse the model about user requirements vs. its own prior responses.

## 5.2 APTITUDE VS. RELIABILITY ANALYSIS

Results presented in Table 1 present averaged performance degradation ($\overline{P}$). We now report on the aptitude and reliability analysis based on metrics $A$ and $U$. Figure 5b visually summarizes the results of the reliability analysis we conducted on the 15 LLMs included in our simulation experiment. First, looking at the two single-turn settings, we see that models that are more able (higher A) tend to be more reliable (lower U). For instance, the two most able models (GPT-4.1 and Gemini 2.5 Pro) achieve the lowest unreliability. At the lower end, the two models with the lowest aptitude (Llama3.1-8B-Instruct and OLMo-2-13B) are also the most unreliable. In summary, **in single-turn settings, models with higher aptitude tend to be more reliable.** This fact is known in the community, with arguments made that better models require less prompt engineering, as they are more robust to variations in inputs and outputs (Li et al., 2023).

The sharded setting paints a different picture. Aptitude degrades in a non-significant way between the full and sharded settings, with an average drop of 16%. On the other hand, unreliability skyrockets with an average increase of 112% (more than doubling). More interestingly, though better models tend to have slightly higher multi-turn aptitude, all models tend to have similar levels of unreliability. In other words, **in multi-turn, underspecified settings, all models we test exhibit very high unreliability, with performance degrading 50 percent points on average between the best and worst simulated run for a fixed instruction.** This refines our definition of the *lost in conversation* phenomenon: when comparing single- and multi-turn settings, we find that large performance degradations ($\overline{P}$) are due in large part to increased model unreliability (U), rather than a loss in aptitude (A).

Appendix F explores potential root causes for models getting lost in conversations. We identify four specific causes: (1) LLMs prematurely propose full answer attempts, making assumptions about problem specifications that lead to confusion (§F.1), (2) they overly rely on previous (incorrect) answer attempts leading to lengthier "bloated" answers (§F.2), (3) LLMs overly adjust their answers based on the first and last turn of conversation, evidenced by a loss-of-middle-turns phenomenon

(§F.3), and (4) they produce overly verbose answers, which likely introduces assumptions that detract attention from user utterances (§F.4).

### 5.3 GRADUAL SHARDING EXPERIMENT

Revealing minimal amounts of information in each turn (Section 3.2) can seem unrealistic and adversarial. To explore the relationship between the granularity of sharding and the severity of the effect, we propose the gradual sharding experiment.

In the gradual sharding experiment, we selected 31 instructions from our original experiment across multiple tasks and expanded each sharded instruction into seven variants, with the shard-set size growing from 2 to 8 shards. The instruction selection and sharding process are detailed in Appendix L. The process ensured that at each shard set size (from 1 to 8), task complexity is fixed, and the only modified factor is sharding granularity.

We ran simulations for the gradual sharding experiments with two models (GPT-4o and GPT-4o-mini), with results summarized in Figure 5c. We find that both models get lost in conversation (a minor degradation in aptitude and a large increase in unreliability) with two-shard instructions and beyond. In other words, the gradual sharding experiment indicates that **any conversation that involves underspecification and occurs in two or more turns leads to models getting lost in conversation**. For users, the granularity at which information is specified does not majorly impact reliability: providing all the information at once (1-shard) is the only effective method to improve reliability.

We note a design decision that affects interpretation: because we selected only instructions with exactly 8 shards, the resulting instructions are inherently more complex than the corpus average (3 to 5 shards). Consequently, even the 2-shard variants in the gradual sharding experiment represent more complex instructions, which may explain why the large drop at $N=2$ is not followed by a proportionally steeper decline as $N$ grows to 8. A complementary experiment using instructions with varying instruction complexity counts would likely yield a more gradual degradation curve, but would confound granularity with task complexity.

## 6 IMPLICATIONS SUMMARY

**System and Agent Builders (Appendix G.1)** Modern LLM applications often rely on agent frameworks like LangChain and Autogen to orchestrate problem decomposition, retrieval, and tool use, raising the question of whether LLMs need native multi-turn capabilities at all. We simulate two agent-style interactions (RECAP and SNOWBALL) that repeatedly inject past user instructions to mitigate the performance drop seen in underspecified multi-turn scenarios. In RECAP, once all shards have been revealed, a final corrective turn informs the assistant that its prior solutions were incorrect, restates all shards as a consolidated bullet list, and asks it to try once more. Both strategies improve over SHARDED but fall short of FULL or CONCAT performance. SNOWBALL offers the more achievable improvement in practice (15–20%) through cumulative repetition. **The findings show that offloading memory to agent frameworks is not sufficient; LLMs should natively support multi-turn interaction.** We also investigate the effect of altering the system prompt, providing an explicit hint to the assistant that the conversation is likely to be multi-turn and underspecified, and found that such a system prompt hint leads to modest gains in performance (+1% across tasks), but does not effectively help the model avoid getting lost in conversation.

**LLM Builders (Appendix G.2).** While the community has focused on improving LLM aptitude, our findings emphasize the importance of model reliability, especially in multi-turn settings. We conducted a temperature ablation experiment (setting $T = 1.0, 0.5, 0.0$) and found that reliability does improve at lower temperatures in single-turn conversations, but not in SHARDED multi-turn ones. Even at $T = 0.0$, multi-turn unreliability remains high (30%) due to cascading effects of early stochastic variation. **In short, lowering temperature does not mitigate unreliability in multi-turn contexts**. We urge LLM builders to jointly optimize for aptitude and reliability, developing models that: (1) maintain similar aptitude in single- and multi-turn settings, (2) demonstrate low unreliability ($U_{10}^{90} < 15$) in multi-turn, and (3) achieve this performance at standard temperature ($T = 1$).

**NLP Practitioners (Appendix G.3).** Our proposed sharding procedure is semi-automated but still requires manual effort (3 hours per 100 samples) to ensure quality. We hypothesize that tasks most vulnerable to multi-turn degradation share three properties: (1) they are generative (not extractive), (2) sufficiently complex such that sharding would yield 3+ shards per instruction, and (3) are non-episodic tasks, with each new shard requiring the modification of the entire solution. For applicable tasks, we encourage researchers to release sharded variants alongside fully specified datasets.

**Conversational System Users (Appendix G.4).** Users should be aware of LLMs' reliability limitations, particularly in multi-turn settings. We offer two practical recommendations. First, "if time allows, try again"—starting a new conversation with the same information often yields better outcomes than persisting with a model that has become lost in conversation. Second, "consolidate before retrying"—since LLMs struggle with information dispersed across multiple turns, consolidating requirements into a single instruction improves both aptitude and reliability. Users can achieve this by asking the LLM to consolidate all user turns into an instruction used in a new conversation. These strategies remain cumbersome workarounds rather than principled solutions, highlighting the need for LLMs that can reliably handle multi-turn conversations without requiring such interventions.

Appendix G shares additional perspective for each of the audiences listed above.

# 7 CONCLUSION

In this work, we conduct a large-scale simulation of single- and multi-turn conversations with LLMs, and find that on a fixed set of tasks, LLM performance degrades significantly in multi-turn, underspecified settings. LLMs get lost in conversation, which materializes as a significant decrease in reliability as models struggle to maintain context across turns, make premature assumptions, and over-rely on their previous responses. Additional experiments reveal that known remediations that work for simpler settings (agent-like concatenation or decreasing temperature) are ineffective in multi-turn settings, and we call on LLM builders to prioritize the reliability of models in multi-turn settings.

# 8 ETHICS STATEMENT

The following section reflects on the limitations of the work we presented. We emphasize that the findings in this paper should be interpreted as the outcome of a controlled underspecification stress test, rather than a direct measurement of LLM behavior in natural dialogue with real users.

A first limitation of our work is the reliance on fully automated simulation. By relying on an LLM to simulate user utterances, we can scale our experiments, including running the same simulation multiple times, which would be cost-prohibitive with real users. However, the simulations we obtain are not representative of natural human-AI conversation. The properties of the sharding process (defined in Appendix C) and of the simulation environment (see Section 3.2) ensure that the simulated conversations follow a rather narrow structure, likely not modeling the full range of conversation dynamics that occur with a large, diverse user population. For example, the simulation process ensures a new shard of information is revealed at each turn, and that the last turn of the conversation has specified all the information needed to complete the task which might not happen with real users. Properties P1, P2, and P5 of the sharding process also restrict the scope of the conversation, as sharded instructions closely match an existing fully-specified instruction, with the high-level intent always identified in the conversation's first turn. The minimal nature of shards is also unrealistic and potentially adversarial, though the gradual sharding experiment finds that different levels of shard granularity lead to similar performance degradations, as soon as conversations occur over two turns or more. Apart from sharding granularity, automatic simulation also lacks the nuance that can occur when a human is involved in conversation, from misunderstandings over terminology, giving up due to frustration with system failures (Wester et al., 2024), or the lack of a feasible end goal for certain conversations (e.g., the user wanting a solution to an unsolved problem). Because of these factors, we believe conducted simulations represent a benign testing ground for LLM multi-turn capabilities. To mitigate the artificiality of the setup, we put several guardrails in place: shards were manually reviewed, merged, and edited to represent natural information splits (Section 3.1), and the user simulator selects shards contextually rather than in a fixed or random order (Section 3.2).

These guardrails ensure a minimum level of conversational realism, though more work is needed to approach fully natural user simulation (Naous et al., 2025; Dou et al., 2025; Mehri et al., 2025; Ross & Andreas, 2025). **Because of the overly simplified conditions of simulation, we believe the degradation observed in experiments is most likely an underestimate of LLM unreliability, and how frequently LLMs get lost in conversation in real-world settings.** The experiments serve as a scalable, low-cost experimental environment for studying LLMs in multi-turn settings. We also note that the scoring mechanism itself favors the sharded setting: because the assistant may produce answer attempts on multiple turns, it can be credited for an early correct guess—whether due to benchmark memorization, lucky inference about missing shards, or lenient evaluation. Despite this structural advantage, multi-turn performance remains substantially below single-turn baselines, confirming that additional answer attempts do not compensate for the reliability loss observed in multi-turn conversations.

A second limitation of our work is the focus on analytical tasks. Although we selected a diverse set of both programming and natural language tasks, we restricted experiments to tasks that involve an analytical solution. This restriction limits the scope of our findings, as we do not establish whether models get lost in conversation on more open-ended tasks, such as creative writing (Chakrabarty et al., 2024). This was a conscious choice: though there has been some progress on creative writing evaluation, it is still an active area of research (Chakrabarty et al., 2025), and we relied on more established tasks and metrics for the initial set of experiments. Determining whether degradation occurs – and if so, identifying the magnitude – on creative tasks is an important direction for future work.

A third limitation of the work is the focus on text-only tasks in the English language. Establishing whether models get lost in conversation in other languages, or in tasks that involve multiple modalities in either user or assistant utterances, could help establish the scope of the degradation observed in LLM multi-turn capabilities.

## 9 REPRODUCIBILITY STATEMENT

We plan to release the code used for simulation, and relevant data such as the sharded instructions and corpus of simulated conversations publicly upon acceptance of the work. We hope the transparency will allow the community to verify and build upon our work, for instance by building future systems that are more reliable in multi-turn settings.

There are two considerations that limit the absolute reproducibility of our experiments. First, a majority of our experiments were conducted with API-based LLMs (closed-source), as they are widely considered the state-of-the-art for LLM at the time of our experiments. API providers are known to deprecate old models over time in favor of newer versions, which likely means running our exact experiments will not be possible once the models are retired. Second, all our experiments involve language models that are probabilistic in nature, leading to statistical variance in the results we produce. Though we took steps to improve the statistical robustness of our results (e.g., running each simulated conversations 10 independent times), scaling up the experiments further might help identify findings that we could not with our achieved experimental budget.

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

## A    RELATED WORK ON UNDERSPECIFICATION

The Background (Section 2) reviews the most directly related prior work, focused on multi-turn evaluation. We now cover other related prior works that have studied underspecification.

Prior work on communication and linguistics has identified underspecification as a common feature of human language (Lappin, 2000; Ferreira, 2008; Frisson, 2009; Pezzelle, 2023).

Understanding how LLMs handle underspecified instructions is crucial for improving conversational capabilities. To this end, Herlihy et al. (2024) identified common response patterns such as hedging, refusal, clarification, and interrogation when underspecified queries are presented to conversational LLM systems, and proposed mechanisms to recover from them. Malaviya et al. (2024) highlighted the importance of supporting context for more accurate and principled evaluation of LLM responses on underspecified queries, and Sarkar et al. (2025) showed that a system that proactively rewrites user instructions to account for underspecification leads to improved LLM responses. Shaikh et al. (2025) studied the degree of grounding (*i.e.*, clarifications and follow-up questions) that LLMs perform in conversation logs and observed that they significantly lack in generating follow-up questions, where humans are 15 times more likely to do so. Chang et al. (2025) hired annotators to manually reproduce fully-specified instructions through a chat interface, and found that the users reveal the entirety of the instruction in 34% of the time, leaving some detail underspecified a majority of the time.

Several works have explored direct tasks to evaluate model ability when dealing with underspecification. Liu et al. (2023) introduced AmbiEnt, a natural language inference benchmark, which revealed that understanding ambiguous statements is still a challenge even to the state-of-the-art LLMs. Wildenburg et al. (2024) created the DUST task, which requires the language model to determine the relative levels of specifications between two sentences, finding that when interpreting underspecified sentences, LMs exhibit little uncertainty. Vijayvargiya et al. (2025) evaluated LLM agents for GitHub issue resolution in an underspecified setting, showing that follow-up interactions to supplement information help improve the resolve rate, but detecting the ambiguities in the instructions remains a challenge.

Prior work has classified different root causes for underspecification. First, task underspecification occurs when humans provide incomplete descriptions of the task at hand, which is prominent in "specification-heavy tasks" (Peng et al., 2023). Second, intent misalignment occurs when the AI fails to understand the user's intent or motivation, and is one of the common sources of user dissatisfaction (Kim et al., 2024b; Terry et al., 2023). Finally, Chaturvedi et al. (2024) discuss location and reference ambiguity, in embodied settings that involve physical spaces such as a Minecraft game.

## B    PRECISE DEFINITION OF SHARDED INSTRUCTIONS

Section 3.1 introduces the concept of sharding at a high level. This Appendix offers a more precise definition by first defining mathematical terminology, and then defining properties that a sharded instruction must satisfy to be considered valid.

Let $q$ refer to a single-turn complex query with intended (i.e., correct) output $Y_q^*$. We refer to the atomic content units (ACU) (Liu et al., 2022) of the query as

$$I(q) = [\mathcal{I}, (c_1, \cdots, c_m)]$$

where $\mathcal{I}$ is the primary intent of the query and $(c_1, \cdots, c_m)$ are the sufficient set of clarifications that specify details of how to compute $Y_q^*$ conditioned on $\mathcal{I}$. For $I(q)$ to be considered *atomic*, any rephrasing of $I(q)$ should produce the same target output. Ie. for all $q'$ s.t. $I(q') = I(q)$, then $Y_q'^* = Y_q^*$.

Given the above definition, the *aim* of the sharding process, for a given query $q$, is to identify the atomic content units $I(q)$ and construct a set of shorter instruction *shards* **s**:

$$q' = [s_1, \cdots s_k] \text{ s.t. } I(q) = I(q')$$

where the shards $s_j$ can be used to simulate multi-turn conversation, with the same intended output as $q$.

A sharded instruction $q'$ is considered valid for an original query $q$ if it fulfills the following properties:

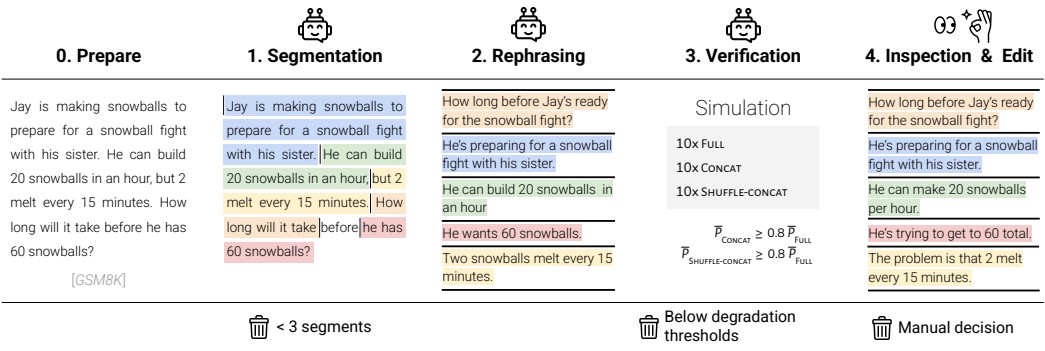

Figure 6: Process diagram of the four-step semi-automatic process to transform fully-specified instructions into a sharded instruction. The first three steps (segmentation, rephrasing, verification) are automated, while the fourth (inspect and edit) was manually completed by the authors of the work. The last row represents the rejection criteria for a sample.

**P1: Information Preservation.** $I(q) = I(q')$ No information from the original instruction necessary for the completion of the instruction should be lost during the sharding process.

**P2: Clear Initial Intent.** $\mathcal{I}_q = \mathcal{I}_{q'}$ and $s_1 = \mathcal{I}_q$. The first shard plays a distinctive role of being the *initial query* within the shard set. The initial query defines the high-level objective for the entire conversation. (e.g., "write a Python function").

**P3: Order Insensitive.** Apart from the first shard, the other shards should be decontextualized (Choi et al., 2021) and not refer to each other in a way that implies an order. As a result, the shard set presented in any order reveals equivalent information. Let $\rho(\mathbf{s}_{2..k})$ refer to a permutation of the shard ordering, then $I(q) = I(\tilde{q}) \ \forall \tilde{q} = [s_1, \rho(\mathbf{s}_{2..k})]$

**P4: Maximal Sharding.** The sharding process should strive to maximize the number of shards extracted from the original instruction (maximize $k$). This can be achieved by producing shards that introduce a single, specific piece of information.

**P5: Minimal Transformation.** The sharded instruction should maintain the instruction language and avoid simplifying, altering, or interpreting elements of the original instruction as much as possible. Apart from modifications to satisfy properties P1-P4, the sharding process should attempt to limit modifications such that the shards ($[s_1, \cdots s_k]$ are semantically similar to the atomic content units $I(q)$.

## C  SEMI-AUTOMATIC SHARDING PROCESS

We rely on a semi-automatic process to transform fully-specified instructions into their sharded equivalents. The process – illustrated in Figure 6 – consists of a sequence of three automated steps (Segmentation, Rephrasing, Verification) followed by a manual step that was conducted by an author of the paper.

We now detail each step of the process, then go over task-specific details we implemented as needed. We note that as part of our open-source release, we provide all the prompts used in the first three LLM-based steps.

**Step 1: Segmentation** Given an original fully-specified instruction (left-most column in Figure 6), the LLM is prompted to extract *segments* of the instructions. Segments are intended to correspond to the atomic content units (defined in Appendix B). In particular, the prompt indicates that segments must not overlap, and that not all words in the original instruction must belong to a segment. Prompts are task-specific and incorporate at least three few-shot examples of segmentation, to allow for the concept of segmentation to be illustrated through examples. At this stage, any instruction that yields fewer than three segments are filtered out and does not proceed to the next stage.

**Step 2: Rephrasing**   Given the original fully-specified instruction and the extracted segments, this stage consists in rewriting each segment to be decontextualized (Choi et al., 2021) and conversational. In other words, dependencies between segments are resolved, and the ordering is changed such that obtained *shards* adhere to properties P2 and P5. In the example above, the fourth segment (highlighted in orange) becomes the first shard as it reveals the overall intent, and light rephrasing occurs in other shards. The rephrasing prompt is task-specific and includes few-shot examples of rephrasing segmented instructions.

**Step 3: Verification**   Steps 1-2 produce a sharded instruction that can be used to simulate SHARDED and CONCAT conversations. To verify the property P1 (Information Preservation) that no information has been lost during segmentation and rephrasing, we conduct preliminary simulations to evaluate the original and sharded instruction side-by-side. Specifically for each pair of the original and the sharded instruction, we simulate ten FULL conversations with the original instruction, ten CONCAT conversations with the sharded instruction (by concatenating the shards), and ten SHUFFLE-CONCAT conversations. SHUFFLE-CONCAT is a variant of the CONCAT simulation in which all shards (except Shard 1) are randomly permuted before being concatenated. This variant can be seen as a more adversarial version of CONCAT, verifying the property P3 (Order Insensitive). For each simulation type, we calculate the averaged performance $\overline{P}$ over ten runs and filter out instructions that are below an acceptable degradation threshold. Specifically, instructions are acceptable if the following conditions are met:

$$\overline{P}_{\text{CONCAT}} \geq 0.8\, \overline{P}_{\text{FULL}}$$
$$\overline{P}_{\text{SHUFFLE-CONCAT}} \geq 0.8\, \overline{P}_{\text{FULL}},$$

where $\overline{P}_{\text{X}}$ denotes the averaged performance of the simulation type X. If more degradation is observed (*i.e.*, below 80%), it indicates a potential loss of information during sharding, or that decontextualization was not implemented accurately.

**Step 4: Inspect and Edit**   Even though the first three steps define the sharding process and implement some level of quality assurance, they do not guarantee the level of quality required for precise and large-scale experiments due to relying on LLM outputs. To obtain high-quality shards, we reserve step 4 for manual inspection and validation. To facilitate the procedure, we developed a web-based annotation interface. In the interface, an annotator can review a pair of fully-specified and sharded instructions, edit, add, or remove individual shards, and decide to accept or reject sharded instructions. Sharded instructions included in our experiments were all manually reviewed by two authors of the work. The amount of editing and filtering required in this final stage varied by task.

Inspecting and editing an auto-generated instruction typically requires 1-3 minutes per instruction, an order of magnitude less than it would require for authors to write the sharded instructions de-novo from a given fully-specified instruction. As part of our open-source release, we provide all the prompts used during sharding, which we hope can facilitate the sharding of additional tasks.

## D   INSPECTION OF SIMULATED SHARDED CONVERSATION

The sharding simulation environment (described in Section 3) relies on LLM components to simulate the user, classify assistant responses, and extract answers from free-text responses. LLM-based components are likely to fail, and we performed an inspection of 200 simulated SHARDED conversations to understand the level of simulation error and the potential effect on estimating the performance of the assistant LLMs due to the error.

For each inspected conversation, we annotated user turns, assistant turns, and the overall conversation with five specific elements.

For user utterances, we annotated whether the utterance revealed exactly the information from one shard in the sharded instruction (`Shard Fully Revealed`). Specifically, we flagged turns that revealed more than one shard, and turns that revealed a shard only partially. We also annotated each user's turn for whether it is appropriately contextualized in the conversation (`Shard Contextualized`). For example, if the previous assistant's turn asked a binary clarification question (yes/no), then proper contextualization would require a Yes/No response to directly address the assistant's response.

| Inspection | All Tasks | 🎛 Actions | 🔧 Code | 🔢 Math | 🗄 Db |
|---|---|---|---|---|---|
| Shard Fully Revealed | 96.0 | 98.3 | 94.9 | 93.4 | 100.0 |
| Shard Contextualized | 98.4 | 98.3 | 98.3 | 98.3 | 98.6 |
| Strategy Accuracy | 95.2 | 94.7 | 95.5 | 95.6 | 94.7 |
| Extraction Success | 97.0 | 100.0 | 93.4 | 98.4 | 100.0 |
| Overall Success | 97.8 | 100.0 | 96.0 | 96.0 | 100.0 |

Table 2: Results of the manual inspection of 100 simulated SHARDED conversations across four tasks: Actions, Code, Math, and Database. The first column aggregates annotation results on the four tasks.

For assistant utterances, we annotated whether the classified strategy was accurate (`Strategy Accuracy`). For example, if the response is labeled as a clarification, we confirm if it poses a clarification question to the user. When assistant utterances were labeled as answer attempts, we further labeled whether the answer extraction step was successful (`Extraction Success`).

Upon completing the inspection of each user and assistance utterance, we assigned a global label to the entire conversation on whether or not the errors that occurred during simulation (if any) affected the overall validity of the simulation. If not, the simulation was marked as successful (`Overall Success`).

We inspected conversations for four tasks: Actions, Code, Math, and Database. The other two (Summary and Data-to-text) are refining tasks that require an answer attempt at each turn, and do not rely on an LLM-based user simulator. As such, they have limited scope for simulation error.

Table 2 summarizes the results of the inspection annotation. Overall, the simulation environment is highly reliable, with roughly 98% of inspected conversations labeled as successful. Some errors occur in each component. With user simulation, a single shard is fully revealed around 96% of the time, and properly contextualized 98% of the time. The processing of assistant responses also leads to errors: the turn strategy classification is only 95% accurate, and extraction of answer attempts has an accuracy of 97%.

Utterance-level errors did not always affect the validity of the overall simulation. In some cases, we observed that the user simulator would correct an error in an early turn, subsequently in the conversation, or that an error in answer extraction on the wrong answer attempt would occur at a turn, but the extraction would be successful later on. In summary, we empirically find that the simulation environment is largely accurate: though some errors occur, large drops of performance in the SHARDED setting (beyond 2%) are not due to errors caused by the simulator. We also observed that the user simulator occasionally produced corrective-style utterances (e.g., "No, I want to sort in XYZ way"), though this behavior was not explicitly encouraged in the simulator prompt and was relatively rare.

## E   CONCRETE EXAMPLE OF LOSS IN APTITUDE VS. RELIABILITY

Let's imagine we are provided with ten instructions ($N = 10$), each FULL and SHARDED. We run simulations with an LLM, simulating 10 conversations per instruction and setting ($M = 10$). Let's assume the LLM achieves an averaged performance ($\overline{P}$) of 90% in the FULL, and 60% in the SHARDED setting.

Finally, let's assume that the FULL performance is achieved by having perfect performance (*i.e.*, success in 10/10 randomly sampled runs) on 9 instructions, and failing on all the sampled simulations of the last, tenth instruction. In other words:

$$S_{ij}^{\text{FULL}} = \begin{cases} 100, & \text{if } i \in \{1, \dots, 9\} \\ 0, & \text{if } i = 10 \end{cases},$$

where $S_{ij}^{\text{FULL}}$ represents the score for $i$-th instruction at $j$-th simulation run. The aptitude ($A$) and unreliability ($U$) of the LLM for the FULL setting is $A = 90\%$ and $U = 0\%$ (*i.e.*, for each instruction, the 10th and 90th percentile scores are equal).

Let's now consider three conditions for the SHARDED setting that all achieve an averaged performance of $\overline{P} = 60\%$. We illustrate the conditions in Figure 7.

**Situation 1: Drop in Aptitude.** The LLM achieves perfect performance on six of the ten instructions:

$$S_{ij}^{\text{SHARDED}} = \begin{cases} 100, & \text{if } i \in \{1, \ldots, 6\} \\ 0, & \text{if } i \in \{7, \ldots, 10\} \end{cases}.$$

In situation 1, $\overline{P} = 60\%$, $A = 60\%$, and $U = 0\%$. The degradation in performance is entirely explained by a decrease in aptitude, while the reliability remains the same.

**Situation 2: Drop in Reliability.** The LLM achieves mixed performance (6-7 perfect scores per instruction) on nine of the 10 instructions:

$$S_{ij}^{\text{SHARDED}} = \begin{cases} 100, & \text{if } 1 \leq i \leq 3, 1 \leq j \leq 6 \\ 100, & \text{if } 4 \leq i \leq 9, 1 \leq j \leq 7 \\ 0, & \text{otherwise} \end{cases}.$$

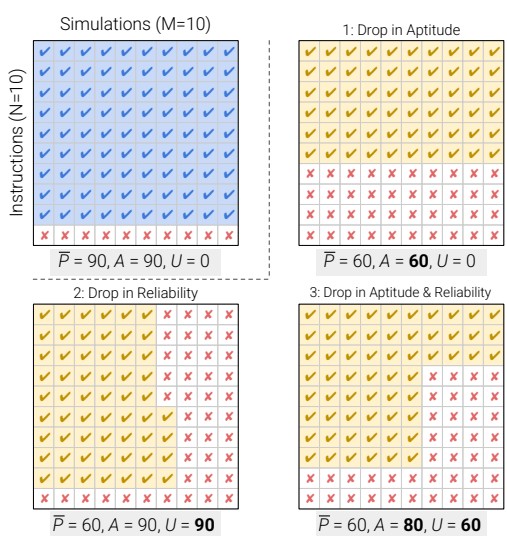

In situation 2, $\overline{P} = 60\%$, with an aptitude of $A = 90\%$, and a unreliability of $U = 90\%$. The degradation in performance is entirely explained by a large drop in reliability, while sharded and fully specified aptitude are equal.

Situations 1 and 2 illustrate extreme scenarios where the average drop in performance is entirely explained by a drop in aptitude or reliability, but in practice, a combination is more likely to occur, as in situation 3.

**Situation 3: Combined drop in Aptitude and Reliability.** The LLM achieves perfect performance on three instructions, and mixed performance (6 perfect scores per instruction) on five of the 10 instructions:

Figure 7: Illustrations for different situations. Green and red fills in each grid indicate sample-level score (*e.g.,* pass / exact match). Compared to FULL (top left), three situations in SHARDED achieve the same $\overline{P} = 60$ while varying in aptitude $A$ and unreliability $U$.

$$S_{ij}^{\text{SHARDED}} = \begin{cases} 100, & \text{if } 1 \leq i \leq 3 \\ 100, & \text{if } 4 \leq i \leq 8, 1 \leq j \leq 6 \\ 0, & \text{otherwise} \end{cases}.$$

In situation 3, $\overline{P} = 60\%$, with an aptitude of $A = 80\%$, and a unreliability of $U = 60\%$. Note that situation 3 leads to a larger increase in unreliability (from 0% to 60%) than a decrease in aptitude (from 90% to 80%) when compared to fully-specific simulations. This corresponds in practice to our observation: drops in performance are explained by small drops in aptitude and large drops in reliability.

Finally, we note that though this concrete example we provide uses binary scores (0 and 100) for simulated conversation outcomes, aptitude (A) and unreliability (U) can equally be applied to continuous metrics (such as BLEU).

## F  QUALITATIVE ANALYSES OF SIMULATION LOGS

In the following subsections, we report qualitative analyses on the corpus of simulations from the main experiment (Section 5.1). The purpose of the analyses is to discern root causes in model

| Model | Conversation Progress At First Answer Attempt | | | | |
|---|---|---|---|---|---|
| | 0-20% | 20-40% | 40-60% | 60-80% | 80-100% |
| *First answer attempt is ...* | earliest | early | midway | late | latest |
| ∞ 3.1-8B | 16.1 | 24.0 | 35.3 | 39.6 | 39.7 |
| OLMo2 | 17.6 | 32.7 | 37.7 | 47.3 | 26.4 |
| A\ 3-Haiku | 27.1 | 35.6 | 47.4 | 58.9 | 70.3 |
| ⑤ 4o-mini | 30.2 | 39.2 | 48.4 | 58.2 | 59.9 |
| ∞ 3.3-70B | 33.3 | 40.1 | 51.2 | 60.0 | 69.3 |
| Phi-4 | 25.7 | 33.1 | 47.0 | 53.0 | 57.9 |
| CMD-A | 38.0 | 42.9 | 56.5 | 65.5 | 73.5 |
| ∞ 4-Scout | 39.8 | 36.8 | 51.0 | 57.9 | 64.8 |
| ⑤ o3 | 21.0 | 37.9 | 51.9 | 58.4 | 68.0 |
| A\ 3.7-Sonnet | 29.2 | 35.6 | 55.3 | 68.0 | 71.6 |
| R1 | 39.5 | 43.1 | 53.5 | 66.4 | 50.2 |
| ⑤ 4o | 36.0 | 41.4 | 56.2 | 65.6 | 90.4 |
| 2.5-Flash | 39.0 | 48.6 | 60.2 | 70.8 | 74.6 |
| ⑤ 4.1 | 33.9 | 52.7 | 60.6 | 69.0 | 78.6 |
| 2.5-Pro | 41.1 | 45.7 | 53.5 | 64.6 | 63.8 |
| Average | 30.9 | 40.5 | 51.7 | 60.4 | 64.4 |

Table 3: Averaged performance ($\overline{P}$) breakdown, based on how early in the conversation the LLM makes its first answer attempt. Analysis conducted on simulations of two tasks: Code and Math.

behavior that lead to performance degradation. We identify four behaviors below and detail the analysis for each in the rest of the section:

1. LLMs attempt to answer the entire problem prematurely.
2. LLMs overly rely on previous (incorrect) answer attempts, leading to lengthier "bloated" answers.
3. LLMs overly adjust their answers based on the last conversation turn, materialized by a pronounced forgetting of middle-turns.
4. LLMs produce overly verbose answers, which likely introduce problem assumptions that detract attention from user utterances.

### F.1 PREMATURE ANSWER ATTEMPTS

During SHARDED simulation, responses are classified according to a seven-class conversation strategy categorization. In particular, each assistant response is tagged as being a formal *answer attempt* or not (as answer attempts require further processing: extraction and evaluation by the task-specific evaluator).

On the onset of conversation, LLMs have the least amount of information (highest level of underspecification) and are least likely to formulate correct answer attempts. Proposing a solution early might therefore plant certain incorrect elements in it, which wrongly influence the interaction later in the conversation.

To evaluate this hypothesis, we bin all simulated conversations from our experiments based on how early in the conversation the first answer attempt is generated by the LLM. Specifically, we create five bins: `0-20%` if the first answer attempt occurs within the first 20% turns of the conversation, and `20-40%`, `40-60%`, `60-80%`, and `80-100%` if it occurs in later turns of the conversation. Of the six tasks included in our experiments, only two (Math and Code) observed a significant range in LLM behavior for answer attempt timing. For the other four tasks, models attempt an answer from the first turn most of the time, rendering analysis on this parameter impossible.

Analysis results for the two remaining tasks are presented in Table 3. We observe that for every single model, conversations with a later first answer attempt lead to higher averaged performance. Across all models, conversations with the first attempt being made in the first 20% of conversations achieve

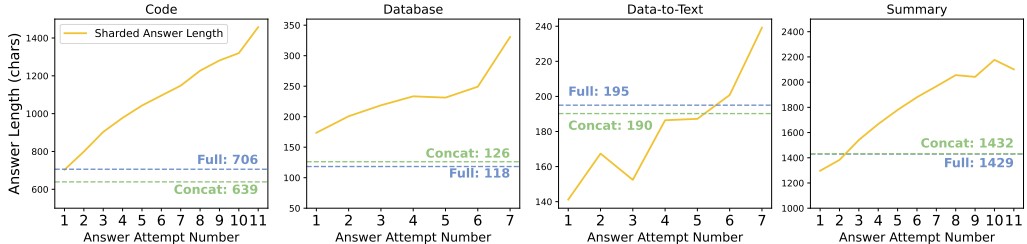

Figure 8: Average length (in number of characters) of answer attempts across four tasks (Code, Database, Data-to-text, and Summary) in SHARDED conversations. Answer attempts in the FULL and CONCAT settings tend to be shorter on average than those from SHARDED setting. SHARDED answer attempts increase in length as the LLMs make more answer attempts.

a score of 30.9, less than half of the 64.4 when the LLM waits for the last 20% of the conversation to make an answer attempt.

In other words, we find that premature answer attempts detract LLM performance. Conversations where the model clarifies user instructions or discusses the problem at a high level before moving to generating complete answer attempts lead to higher performance. We hypothesize that this is due to the model making incorrect assumptions in premature solutions, which conflict with subsequent user instructions in later turns.

### F.2 ANSWER BLOAT IN MULTI-TURN CONVERSATION

In multi-turn conversation simulations, the LLM might make multiple answer attempts, with each subsequent attempt being potentially based on previous attempts. In contrast, single-turn conversations constrain conversation dynamics, with the LLM making a single, first-and-final answer attempt.

To understand multi-turn conversation dynamics, we calculate the average length of answer attempts in each simulation type. For the SHARDED setting, we calculate average length for each attempt within a simulation (*i.e.*, average length of the first attempt, second attempt, third attempt, etc.). We note for readers here that the analysis is conducted on extracted answer attempts (output of the Answer Extractor module in Figure 2) rather than the entire assistant responses. The extracted answer more accurately measures dynamics in answer attempts (i.e., generated SQL query, or Python function) rather than the entire responses, which might contain varying amounts of unrelated content.

Results of the analysis are plotted in Figure 8. Across the four tasks, we find that answer lengths in the FULL and CONCAT settings tend to be similar, typically within 2-10% of each other. On three of the analyzed tasks (Code, Database, Summary), the first answer attempt in the SHARDED setting has a similar length to FULL and CONCAT counterparts, yet for each subsequent answer attempt, we observe an increase in average answer length. The effect is such that the final answer attempts in SHARDED conversations (right portion of the four plots) tend to be 20-300% longer than the solutions generated in the FULL and CONCAT settings. We name this observation the *answer bloat effect*: as a multi-turn conversation progresses, the LLM generates incorrect answer attempts, making assumptions about portions of the instruction that remain unspecified. As the user reveals additional information in succeeding turns, the LLM does not successfully invalidate its prior assumptions and overly relies on its previous attempts. Answer bloat in multi-turn, underspecified conversation leads to longer solutions compared to single-turn equivalents.

We perform an additional analysis, focusing only on the Code and Database tasks and filtering to simulations where the LLM reaches an entirely correct solution (score of 100.0). For Code task, correct programs obtained from SHARDED setting are on average 850 characters long, which is 27% more characters than the correct solutions generated in the FULL setting (668 characters on average). For Database, correct SQL queries in the SHARDED setting are on average 129 characters, 14% more characters than those from the FULL setting (113 characters). In summary, LLMs are less likely to reach a correct solution in multi-turn settings (lower $\overline{P}$), and when they do, the final solutions they reach are longer (bloated), hinting that the solutions are qualitatively worse.

### F.3 OVER-ADJUST BASED ON LAST TURN OF CONVERSATION

Because the summary task requires the assistant to attribute its summary back to documents through citation, the task offers a unique opportunity to analyze what turns of information LLMs pay attention to as the multi-turn conversation progresses. As a reminder, the summary task involves a user introducing new documents at each turn. Therefore, our analysis aims to understand whether document introduction order (across turns) affects the likelihood of the LLM citing a document.

In Figure 9, we plot the results of our analysis. Each row corresponds to the analysis of summaries generated at a given turn in the sharded simulation. At turn 1 (top row), 96% of the cited documents were introduced in the first turn. The missing 4% corresponds to hallucinated citations to documents that were not introduced, and explains why none of the rows' distributions sum to 100%. At turn two (second row from the top), summaries include citation in roughly equal proportion for turn-1 and turn-2 documents (i.e., 48% and 49%).

We interpret this to mean that in 2-turn conversations, LLMs pay roughly equal attention to documents in either turn. Analysis of summaries generated in turns 3-8 of sharded simulations reveals an imbalance in the documents the LLM cites to. In eighth-turn summaries, 20% of citations are to documents introduced in turn 8, compared to 8% from turn 2 and 3 (150% difference). At a high level, as the conversation progresses, LLMs are most likely to cite either documents in the first or last turns, and less likely to cite documents introduced in intermediary (middle) turns. This finding mirrors findings of a *loss-in-the-middle* phenomenon of LLMs paying more attention to documents at the start or end of their provided context, at the cost of middle-context content (Huang et al., 2023; Liu et al., 2024; Laban et al., 2024). In short, we observe that the lost-in-the-middle phenomenon occurs not only in single-turn long-context settings but also in multi-turn conversations. We name this phenomenon *loss-in-middle-turns*.

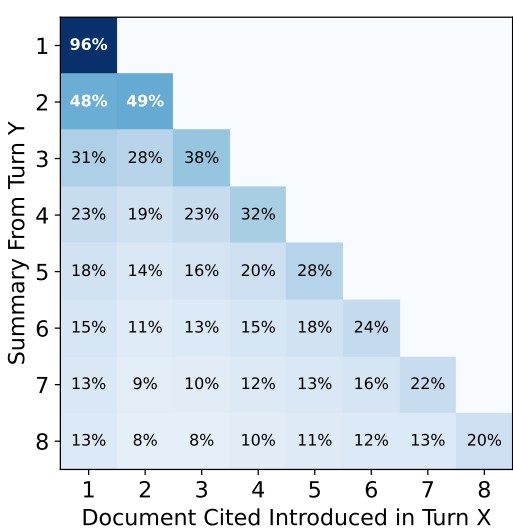

Figure 9: Analysis of citation patterns in summaries generated by LLMs with the SHARDED simulation. At each turn, the LLM generates an updated summary (y-axis), which includes citations from the documents that have been revealed up to this turn. Percentages in a row do not add up to 100% due to citation hallucinations that occur for some models.

We note that the analysis presented in Figure 9 averages numbers across the 15 LLMs included in our main experiment. Even though we observe some loss-in-middle-turns in all models, the magnitude of the effect varies across models, typically with more performant models having a more muted effect, showing they have better capabilities of handling attribution across multiple turns of context. We do not include model-specific analyses in this work and leave it for future work.

### F.4 OVERLY-VERBOSE ASSISTANT RESPONSES

When simulating multiple conversations based on a common instruction, we observe variation in responses, particularly in the length of the response generated by the LLM. To understand how verbosity (length of a response) affects model performance, we perform a verbosity analysis.

One difficulty with assessing verbosity is that different tasks and instructions might require different levels of verbosity. For example, generating a Python function likely requires a longer than generating an SQL query. To regularize for task-specific variations, we assign a *verbosity tag* calculated for each (LLM, instruction) tuple. For each simulated sharded conversation involving an LLM on an instruction, we calculate the average length of the per-turn response (number of total characters in assistant responses divided by number of turns). We then bin conversations into quintiles according to this metric. More specifically, since we simulated $N = 10$ conversations for each (model, instruction)

| Task | Relative Assistant Verbosity | | | | |
|------|--------|--------|--------|--------|---------|
| | 0-20% | 20-40% | 40-60% | 60-80% | 80-100% |
| *Assistants responses are ...* | shortest | short | median | long | longest |
| Code | 55.3 | 52.3 | 48.9 | 46.9 | 42.5 |
| Math | 62.9 | 64.0 | 62.1 | 60.9 | 56.1 |
| Database | 43.8 | 40.0 | 37.3 | 34.3 | 31.3 |
| Actions | 41.5 | 49.6 | 54.2 | 53.6 | 50.8 |
| Data-to-Text | 25.0 | 24.3 | 24.0 | 23.1 | 21.8 |
| Summary | 15.4 | 14.7 | 13.5 | 12.0 | 10.3 |
| Average | 40.7 | 40.8 | 40.1 | 38.6 | 35.6 |

Table 4: Averaged performance ($\overline{P}$) of LLMs on the six experimental tasks, arranged based on model relative verbosity (length of response). Performance degrades when models generate longer responses on five of the six tasks.

pair, we assign 2 simulations per quintile, which we name: shortest, short, median, long, and longest. We then calculate the averaged performance ($\overline{P}$) on the six experimental tasks, arranged based on this verbosity tag. Results are summarized in Table 4.

On five of the six tasks, performance is 10-50% higher in simulated conversations with the shortest response length, compared to conversations with the longest response length. As assistant responses get longer (left to right in the Table), performances gradually drop. The Actions task is the only task where such an effect is not observed, and where the shortest response length from the assistant is detrimental.

However, models primarily achieve higher performance when they generate shorter responses. We hypothesize that deterioration due to over-verbosity is due to longer responses typically containing more assumptions or hypotheses from the assistant, which can lead to confusion in following turns. On the other hand, short turns tend to be focused (e.g., a single clarification question) and keep the conversation on track.

Deterioration due to over-verbosity is noteworthy, as besides deteriorating underlying model performance, longer responses also take longer for users to read, which is undesirable. Therefore, the finding indicates that longer LLM responses are bad both for models and end-users.

# G    IMPLICATIONS

## G.1    IMPLICATIONS FOR SYSTEM AND AGENT BUILDERS

Building LLM-based applications typically involves complex processes: decomposition of problems, retrieval of relevant information, use of tools, and calling of actions. Such processes are typically orchestrated by an agentic framework (such as Autogen (Wu et al., 2023) or LangChain (Chase, 2022)) that allows system builders to compose workflows with LLM calls as individual blocks. As such, an argument could be made that multi-turn capabilities are not a necessary feature of LLMs, as it can be offloaded to the agent framework. In other words, do we need native multi-turn support in LLMs when an agent framework can orchestrate interactions with users and leverage LLMs only as single-turn operators?

To answer this question, we implemented two agent-style conversation simulation types: ⟳ RECAP and ☺ SNOWBALL. Both preprocess user utterances before sending them to the LLM. In RECAP, a conversation proceeds in the same way as SHARDED, but a user turn is added at the end, which recapitulates all the previous user turns. SNOWBALL is a more gradual recapitulation: at each turn, the user simulator reveals a new shard, and repeats all previously revealed shards at that point. Both simulation types repeat the past user's turn information to make it more prominent and give the LLM a chance to leverage the redundancy to improve its responses. We include the experimental detail in Appendix N.

| Model | Simulation Type | | | | |
|---|---|---|---|---|---|
| | 📄 | 🗄 | 🪷 | 🔁 | 😊 |
| 🔷 4o-mini | 86.8 | 84.4 | 50.4 | 66.5 | 61.8 |
| 🔷 4o | 93.0 | 90.9 | 59.1 | 76.6 | 65.3 |

Table 5: Experimental Results with additional simulation types: 🔁 Recap and 😊 Snowball. Both strategies involve repeating user-turn information to mitigate models getting lost in conversations.

| Model | Tasks | | | | Avg. |
|---|---|---|---|---|---|
| | 🖩 | 🤖 | 🗄 | ➕ | |
| 🔷 4o | 67.9 | **65.0** | 42.3 | 61.3 | 59.1 |
| 🔷 4o + system prompt | **68.7** | 59.0 | **45.2** | **68.0** | **60.2** |

Table 6: Comparing performance in SHARDED conversations of GPT-4o given no system prompt (default) vs. providing a specialized system prompt hinting that the conversation will likely be underspecified. Results reported on four tasks: 🖩 Math, 🤖 Actions, 🗄 Database, and ➕ Code.

Table 5 summarizes the results on all instructions for four tasks (Code, Database, Math, Actions) for two tested models (GPT-4o, GPT-4o-mini). Both RECAP and SNOWBALL demonstrate some level of success, with improvements over SHARDED simulations, but the performance still lags behind FULL or CONCAT. While RECAP outperforms SNOWBALL, we note that RECAP is an unrealistic setting because the intervention is conducted on the *last turn* of the conversation, which is not known a priori when conversation unfolds with a real user. SNOWBALL gives a sense of realistic performance gains achievable through user-turn repetition: it can mitigate the FULL-to-SHARDED performance deterioration by 15-20%. In short, relying on an agent-like framework to process information might be limiting, and we argue LLMs should natively support multi-turn interaction.

Besides agent-like procedures such as snowball and recap, we conducted an additional experiment to study the effect of system prompts on assistant model performance. Specifically, we re-ran sharded simulations with the GPT-4o adding an additional sentence at the end of the system prompt provided to the assistant during simulation: "*Be aware that the user might not provide all the information needed to solve the task at once, and that you can ask for clarifications if needed.*" The intent of this additional system specification is to observe the model can shift its behavior based on the provided hint, and effectively avoid getting lost in conversation.

Results of the experiments on the four non-refinement tasks are summarized in Table 6. In summary, adding the system prompt yields small improvements (+1 to +3) on two tasks, large improvements on one task (+7) and a drop on one task (-6), totaling a small improvement (+1%) on average. In conclusion, we do observe qualitatively that adding a hint to the system prompt affects model behavior, but it does not successfully help the model avoid getting lost in the conversation.

## G.2 IMPLICATIONS FOR LLM BUILDERS

A lot of effort has been put in improving LLM *aptitude*: demonstrating that LLMs can accomplish tasks of increasing intellectual complexity, with recent results showing LLMs can compete in mathematics Olympiads, or solve Ph.D.-level technical questions in a benchmark aptly named Humanity's Last Exam (Phan et al., 2025).

In this work, we call on LLM builders to prioritize *reliability* of the models they build, as our experiments demonstrate that the randomness involved in generating text with LLMs leads to catastrophic unreliability in all the models we tested, degrading the quality of responses the average LLM users see.

LLMs are probabilistic systems, with parameters such as *temperature* that can adjust the degree of randomness that occurs while generating text. A possible argument is therefore: does setting the temperature to its lowest setting ($T = 0$) effectively resolve the reliability concern, as it makes the generation process more (but not entirely) deterministic?

| Simulation | 🦙 4o-mini | | | 🦙 4o | | |
|---|---|---|---|---|---|---|
| | AT=1.0 | AT=0.5 | AT=0.0 | AT=1.0 | AT=0.5 | AT=0.0 |
| 📄 FULL | 16.0 | 15.0 | 6.8 | 17.8 | 8.0 | 2.8 |
| 🗄 CONCAT | 20.2 | 17.8 | 9.5 | 20.2 | 17.8 | 5.8 |
| 🦐 UT=1.0 | 49.8 | 46.8 | 51.0 | 41.0 | 43.8 | 31.8 |
| 🦐 UT=0.5 | 31.7 | 34.0 | 40.5 | 39.5 | 40.8 | 31.8 |
| 🦐 UT=0.0 | 38.5 | 28.0 | 30.5 | 35.8 | 38.0 | 29.7 |

Table 7: Unreliability of models when changing assistant temperature (AT) and user temperature (UT) in 📄 FULL, 🗄 CONCAT and 🦐 SHARDED settings. The lower the number the more reliable the assistant is.

To evaluate this argument, we conducted a supplementary experiment in which the assistant's temperature for generating responses (AT) was varied to three values: 1.0, 0.5, and 0.0. Additionally, since SHARDED simulation uses an LLM-based user simulator, we also varied the user's temperature (UT) with the same three values. Further details on the experiment, including sample selection and simulation scale, are in Appendix M.

Table 7 summarizes the experimental findings. Looking at the FULL and CONCAT settings (first two rows), both GPT-4o-mini and GPT-4o observe a large improvement in reliability when temperature is decreased, with a drop in unreliability ($U_{10}^{90}$) of 50-80% when the assistant temperature decreases. Results from SHARDED simulations are more alarming: GPT-4o-mini does not see improvements in reliability as AT is decreased (in all user-temperature settings), and GPT-4o only sees minor improvements, on the order of 15-20%. Even when both the user and assistant temperatures are set to 0.0, there remains a large unreliability of around 30%. Even though language models are supposed to be deterministic at $T = 0.0$, this is known to practically not be the case for modern LLMs (see Appendix O for discussion). At a high level, single-turn conversations have limited scope for deviation, whereas one token difference in an early turn of a multi-turn conversation can lead to cascading deviations, which we observe as stagnated unreliability. For settings that involve multi-turn interaction, we find that **lowering the temperature of the LLM when generating responses is ineffective in improving system reliability.**

We invite and challenge LLM builders to jointly optimize model aptitude and reliability. A reliable LLM should: (1) achieve similar aptitude in single- and multi-turn settings, (2) have small unreliability ($U_{10}^{90} < 15$) in multi-turn settings, (3) achieve these at unmodified temperature ($T = 1.0$), demonstrating that the underlying language model can handle variations that naturally occur in language generation.

### G.3 IMPLICATIONS FOR NLP PRACTITIONERS

Our experiments demonstrate that model behavior in single- and multi-turn settings on the same underlying set of instructions can diverge in important ways, for example, with large observed degradations in performance and reliability.

We selected the initial six tasks to span a wide range of generation tasks, from programming to multi-document summarization. Yet this set of tasks is limited across multiple dimensions, such as focusing on English-language instructions and analytical (i.e., non-creative) tasks. We put effort into making the sharding process scalable by automating portions that could be handled by an LLM, while manually validating and finalizing samples for quality control. The sharding process – detailed in Appendix C – required an average of three hours of manual work (prompt engineering or inspection) from an author to prepare and finalize 100 sharded instructions.

We encourage NLP practitioners to experiment with sharding and release sharded versions of their tasks and instructions alongside fully specified ones.

To illustrate the feasibility of sharding new tasks, and understand compatibility requirements for sharding, we prepared sharded instructions for a seventh task: 🔤 Translation. The task consists of translating an entire document (10 sentences) from German to English, leveraging paired documents from WMT 2019 on document-level translation (Scherrer et al., 2019). In the SHARDED setting, each turn reveals two additional sentences from the source document and requires the assistant to translate

| Model | 🔤 Translation | | |
|-------|:-----:|:-----:|:-----:|
|       | 📄 | 🗄 | ⚒ |
| 🌀 4o-mini | 41.7 | 43.4 | 42.1 |
| 🌀 4o | 35.9 | 38.5 | 40.9 |

Table 8: Performance on the 🔤 translation task for 📄 FULL, 🗄 CONCAT, and ⚒ SHARDED simulations.

all sentences provided so far, whereas the FULL and CONCAT settings reveal the entire document in the first turn. Evaluation is conducted with the standard BLEU metric (Papineni et al., 2002). We describe practical implementation details in Appendix J.

Results from FULL, CONCAT, and SHARDED simulations are summarized in Table 8. Both models we tested – GPT-4o-mini and GPT-4o – do *not* exhibit degradation in performance in the SHARDED setting, with BLEU scores being within 10% difference of each other in all settings. We believe this result reflects that the task can largely be accomplished at the sentence-level despite some prior work that has framed translation at the document-level (Post & Junczys-Dowmunt, 2023), and that the BLEU score does not adequately capture document-level nuances (Ma et al., 2021). In other words, if a task is episodic (i.e., it can be decomposed into turn-level subtasks), the models can avoid getting lost in conversation by completing each subtask without having to handle multi-turn context. In short, the SHARDED Translation task simulates multi-turn conversations that are not underspecified.

We now list task properties we believe are important in leading models to get lost in conversation in multi-turn settings. First, generative tasks (*i.e.*, unlike extractive QA or classification) are more prone to model confusion, as they typically involve editing and refinement of new content. Second, the generative tasks should be sufficiently complex, involving multiple explicit specifications that will yield a multitude of shards. For example, an instruction: "Write a Python program that calculates $1 + 1$" is too simple to shard. Third, the solution or answer should be non-decomposable, such that revealing a shard modifies the entire solution (unlike the translation task, where each additional shard only asks to translate and append to the ongoing solution). We hypothesize that LLMs tested on tasks with the aforementioned three properties will likely get lost in conversation, evidenced by a large drop in averaged performance and reliability in SHARDED simulations.

### G.4 IMPLICATIONS FOR USERS OF CONVERSATIONAL SYSTEMS

Users of LLM-based products should be aware of the lack of reliability of LLMs, particularly when used in multi-turn settings. Generally available generative technology is new, and prior work has identified the randomness in LLM-generated text as a point of confusion for users (Mylrea & Robinson, 2023; Weisz et al., 2024; Venkit et al., 2024; Lee et al., 2024). We make two practical recommendations that can help users of LLM-based systems get the most out of their exchanges.

**If time allows, try again.** If a conversation with an LLM did not lead to expected outcomes, starting a new conversation that repeats the same information might yield significantly better outcomes than continuing an ongoing conversation. This is because current LLMs can get lost in the conversation, and our experiments show that persisting in a conversation with the model is ineffective. In addition, since LLMs generate text with randomness, a new conversation may lead to improved outcomes.

**Consolidate before retrying.** Since LLMs are ineffective at dealing with information dispersed across multiple turns, consolidating instruction requirements into a single instruction is an effective strategy to improve the model's aptitude and reliability (as shown by the CONCAT experiments). When a user notices that a model is lost in conversation, they can ask the LLM: "Please consolidate everything I've told you so far," then bring the response to a new conversation, alleviating the need for manual consolidation. In practice, there is anecdotal evidence that early adopters of LLM-based applications are aware that LLMs get lost in conversation. For example, users of the Cursor LLM-based coding environment report that frequently creating new conversations "whenever they can"

| Name | Description | Example |
|---|---|---|
| Answer attempt | The response contains a complete answer attempt to the question that can be extracted verbatim. | The dog is 50 meters away from the house. |
| Clarification | The response is a brief single question that directly inquires about one aspect of the query. | To calculate the distance, I need to know how long the dog ran. Could you provide more information about that? |
| Interrogation | The response contains multiple questions addressed to the user. | I cannot answer the question without knowing (1) speed, (2) duration, and (3) starting position. Please tell me about these points and I can calculate the distance! |
| Discussion | The response discusses the question in detail without answering, asking, or refusing to answer. | The question is trying to measure the distance between the dog and the house. We can calculate based on this equation: [Equation]. [. . .] |
| Hedging | The response provides multiple answer candidates based on hypotheticals (ifs, cases). | 1. If the dog was originally in the house, it would be 50 meters away now. 2. If the dog was at the park, it would be 100 meters away from the house now. |
| Refusal | The response refuses to answer the question without a follow-up question or a request. | I can't answer your question because I don't have sufficient information. |
| Missing | The response is empty. | [blank] |

Table 9: Definition of turn categories. We include the description in the prompt to categorize assistant responses.

is a recommended strategy to ensure high quality responses even though the tool allows to keep conversations going indefinitely.[6]

These two recommendations remain cumbersome for users and can only offer patched solutions rather than a principled approach. Once future LLMs can more reliably handle multi-turn conversations, the need for such recommendations should be alleviated, allowing users to communicate underspecified instructions over multiple turns naturally with less risk of the model getting lost in conversation.

## H ASSISTANT RESPONSE CATEGORIZATION

We categorize each assistant response into one of the seven categories to capture the answer attempt and evaluate if that is the case, as well as to understand the model's behavior tendency. Herlihy et al. (2024) defined seven turn categories for LLM responses and classified them using LLM, uncovering that GPT-4 prefers answering directly even when the query is underspecified. Motivated by this study, we similarly define seven response categories, which we list in Table 9, together with example responses. Key differences are discussion and answer attempt; we observed many responses containing a large body of text formulating the question in our preliminary experiments, which led to redefining "Miscellaneous" from (Herlihy et al., 2024) into "Discussion" in our experiment. "Direct Response" in (Herlihy et al., 2024) corresponds to our "Answer Attempt."

## I MODEL ACCESS

We accessed models that were used in the experiments from various vendors. The short form names we used throughout the paper, the corresponding versions, and the providers are summarized in Table 10. Except for the exploration with various temperatures (Section G.2), we set the temperature to $T = 1.0$ and used the default values for the rest of the configurable hyperparameters. We set the maximum response length to 1,000 tokens for all models, and did not observe models exceeding this

---

[6]https://www.reddit.com/r/cursor/comments/1j72r8d/when_to_start_a_new_chat/

limit frequently when generating responses. For thinking models (o3, Deepseek-R1), we increased the limit to 10,000 tokens to account for the additional test-time compute (thinking tokens).

| Short Form | Name | Version | Access Provider |
|---|---|---|---|
| 4o | GPT-4o | `gpt-4o-2024-11-20` | OpenAI / Microsoft API |
| 4o-mini | GPT-4o-mini | `gpt-4o-mini-2024-07-18` | OpenAI API |
| 4.1 | GPT-4.1 | `gpt-4.1-2025-04-14` | OpenAI / Microsoft API |
| o3 | o3 | `o3-2025-04-16` | OpenAI / Microsoft API |
| 3-Haiku | Claude 3 Haiku | `claude-3-haiku-20240307` | Amazon Bedrock |
| 3.7-Sonnet | Claude 3.7 Sonnet | `claude-3-7-sonnet-20250219` | Amazon Bedrock |
| 2.5-Flash | Gemini 2.5 Flash | `gemini-2.5-flash-preview-04-17` | Gemini API |
| 2.5-Pro | Gemini 2.5 Pro | `gemini-2.5-pro-preview-03-25` | Gemini API |
| 3.1-8B | Llama-3.1-8B-Instruct | N/A | Local Ollama |
| 3.3-70B | Llama-3.3-70B-Instruct | N/A | Amazon Bedrock |
| 4-Scout | Llama-4-Scout-17B-16E | N/A | Together AI |
| CMD-A | Command-A | `command-a-03-2025` | Cohere API |
| R1 | Deepseek-R1 | N/A | Amazon Bedrock |
| OLMo2 | OLMo2-13B | N/A | Local Ollama |
| Phi-4 | Phi-4 | N/A | Local Ollama |

Table 10: Specific model versions used as part of our experiments. For each model, we define the exact `Version` of the model accessed (for models that have versioning) and the `Access Provider` to facilitate result reproducibility.

## J    TASK-SPECIFIC IMPLEMENTATION DETAILS

We provide task implementation details. For each task, we specify: (1) the selection of original single-turn fully-specified instruction, (2) the evaluation metric that was repurposed from the original dataset, and (3) what the initial system messages consist of (if any).

### J.1    CODE

The Code instructions are sourced from a combination of HumanEval (Chen et al., 2021), a dataset of 164 basic Python programming problems given the function header and the docstring that specifies the problem, and LiveCodeBench (Jain et al., 2024), an evolving dataset of Python algorithmic challenges. In particular, we source from the "call-based" problem subset in LiveCodeBench v5, with the difficulty of either "Easy" and "Medium", to align the solution formats between the two sources.

We first sharded all HumanEval problems following the protocol mentioned in Appendix C, obtaining 45 high-quality sets of shards that meet the criteria. The rest of the dataset was discarded because it was simplistic and did not produce shared instructions with enough shards. Subsequently, we shuffled and sharded the aforementioned subset from LiveCodeBench until obtaining 100 valid sharded instructions.

We follow the original prompts used by the benchmark authors as much as possible for the single-turn (FULL and CONCAT) evaluation. Specifically, FULL prompt from HumanEval includes the function header and the docstring provided as `prompt` in HumanEval dataset, and FULL & CONCAT from LiveCodeBench includes `starter_code` consisting of the function signature.

Both HumanEval- and LiveCodeBench-derived problems come with test cases, which we use to compute the functional accuracy of the answer attempt by the LLMs. We re-use the evaluation codebase maintained by Jain et al. (2024), which (1) wraps the candidate function in a test module, (2) execute given the inputs, and (3) checks the equivalence of the output from the expected output, with a default timeout set to prevent the evaluator from getting trapped during evaluation (*e.g.*, brute-force implementation may not pass under the set time budget). When multiple code blocks are present in a response, the answer extraction module selects the last function definition in the last markdown code block.

## J.2  🗄 DATABASE

The Database instructions are sourced from the validation portion of the Spider dataset (Yu et al., 2018). We note that though a more recent version of Spider has been released (Spider 2.0 (Lei et al., 2024)), the instructions in the second iteration are more advanced and represent less typical database use, and we select instructions from the more realistic Spider 1.0.

The authors of Spider categorized queries into four levels of difficulty (EASY, MEDIUM, HARD, XHARD), based on the syntax complexity of a reference SQL query. We filtered out queries of EASY complexity, as they tended to yield fewer than three shards when processed. The rest of the 433 natural language queries in Spider were gradually sharded until reaching a total of 107 valid sharded instructions.

Each original instruction in Spider supplies a database schema, represented in SQL as a series of table schema (i.e., each defines a series of columns including name, type, and optional index). We include the database schema as part of the system message (i.e., before the first turn of conversation), and inform the LLM that users will provide natural-language queries that must be answered using a database with the provided schema.

Each original instruction in Spider is paired with a reference SQL solution. We follow Zhong et al. (2020) for the evaluation methodology. For a given original instruction, the candidate and reference SQL queries are executed on a fixed set of databases, and an exact match of the results on all databases is required to mark the candidate as successful (Score = 100). If a discrepancy is observed on any test database, the candidate is incorrect (Score = 0). One limitation of SQL execution is that false positives can occur: two queries can return the same output on a given database, even when they are not semantically equivalent. Zhong et al. (2020) found that by evaluating on an increased number of databases, false positives become negligible. Finally, any invalid candidate that does not successfully execute (e.g., syntax error) is considered incorrect (Score = 0).

## J.3  🔧 ACTIONS

The Actions instructions are sourced from the released test portion of the Berkeley Function Calling Leaderboard V3 (BFCL) (Yan et al., 2024). BFCL V3 consists of three sub-genres of instructions: (1) Parallel, (2) Multiple, and (3) Multiple-Parallel. Initial experimentation with the sub-genres identified Parallel as the most suited for sharding, as Parallel instructions specify multiple subtasks that should be used and combined into a single action that accomplishes the entirety of the instruction. We shuffled all the BFCL V3 Parallel instructions, and sharded gradually until we obtained 105 valid sharded instructions.

We note that though a more recent iteration of BFCL includes multi-turn instructions, it differs from sharding experiments as it does not involve underspecification, with each turn having an independent intermediate solution (which we call episodic multi-turn conversations). Our implementation in comparison shards original instructions, allowing us to simulate multi-turn underspecified conversations for this task setting. The Background section (Section 2) discusses the relationship between episodic and underspecified multi-turn conversation more in-depth.

Each instruction in BFCL comes with tool set documentation, a JSON object that specifies the set of available actions (APIs) for the assistant to complete user instructions. We include the tool set documentation as part of the system message, along with a message indicating that user queries will require the use of the provided tools to be completed.

Each instruction in BFCL comes with a reference answer, consisting of the API calls that should be called to accomplish the user instruction. The maintainers of BFCL have released an evaluation toolkit that assesses semantic equivalence between a candidate answer and the reference answer. We leverage the official evaluation toolkit, assigning a score of S=100 for candidate answers that are considered semantically equivalent to the reference answer, and a score of S=0 otherwise. When the evaluation toolkit is not able to parse a candidate answer (e.g., a syntax error), the candidate is considered incorrect (S=0).

### J.4 ⊞ MATH

The Math instructions are sourced from the "main" portion of the GSM8K dataset (Cobbe et al., 2021). We did not perform a filter on the original 8,700 instructions. We shuffled the instructions and sharded incrementally until we obtained 103 valid sharded instructions. Each GSM8K is paired with a numerical reference answer. We used the official toolkit released alongside GSM8K to standardize numerical answers (i.e., strip formatting, etc.). Standardized candidate numerical answers can then be compared through exact match to the reference answer. If the toolkit detects a match, the candidate answer is considered correct (Score=100), and incorrect otherwise (Score = 0). A short, single-sentence system prompt is used to indicate to the assistant that it will be solving mathematical problems.

### J.5 ⊞ DATA-TO-TEXT

The Data-to-Text instructions are based on instructions in the released test set ToTTo dataset (Parikh et al., 2020). In ToTTo, fully-specified instructions have the following information elements: (1) a HTML-formatted table extracted from a Wikipedia page, (2) a subset of cells in the table that have been *highlighted*, (3) the name of the Wikipedia page that included the Table, (4) the name of the Section in the Wikipedia page that included the Table. Given these elements, the task objective is to generate a caption for the Table, specifically focusing on the highlighted cells and considering the available metadata. Instructions were shuffled and sharded incrementally until we obtained 120 valid sharded instructions.

For each instruction, we generate sharded instructions by assigning different information elements to individual shards. The first shard consists of the initial HTML-formatted table without highlighting. The second shard provides an updated table with the highlighting present, the third shard provides the Wikipedia page name, the fourth shard provides the Wikipedia Section name. Finally, a fifth shard provides a fixed set of 10 randomly-selected example captions from the training set of the ToTTo dataset.

Each instruction in ToTTo is assigned one to three reference captions that were collected by the authors of the original dataset. Evaluation on a candidate caption calculates the BLEU score (Papineni et al., 2002) between the candidate and the set of available references, following the evaluation methodology from the original paper.

The Data-to-Text is a refinement task; at each turn, the model is provided an additional shard of information, and is explicitly told to update its response considering all the information provided so far. As a refinement task, assistant responses at each turn are automatically categorized as answer attempts, and the extracted answer is considered to be the entire response. The system instruction informs the model that its response should consist solely of a table caption, without additional text (such as intro, outro, or politeness wording).

### J.6 ⨭ SUMMARY

The Summary instructions are based on samples of the Summary of a Haystack dataset (Laban et al., 2024). We reuse the entire instructions from Summary of a Haystack to produce 92 sharded instructions. The original instructions each consist of a *haystack* – 100 documents for a total of 100,000 tokens of content – and a user query. The goal of the task is to generate a bullet-point-formatted summary of the query-relevant insights that occur in the collection of documents, and use citation to attribute information in each of the bullet points back to the source documents.

The original setting of the Summary of a Haystack purposefully includes a large amount of redundancy (each insight is repeated across at least 6 documents) to evaluate LLMs' ability to thoroughly cite sources. However, we simplify the task for the multi-turn setting, as the 100,000-token haystacks restrict the variety of models we can evaluate. We instead follow subsequent work in selecting smaller Haystacks ("mini-Haystacks") (Belem et al., 2024). Mini-Haystacks consist of 20 documents and ensure that each reference insight is repeated across three documents. For each instruction, we produce ten shards by randomly assigning two documents per shard. The initial shard further specifies high-level task instruction, by specifying the user query, the expected bullet-point format, with a formatted citation.

Summary of a Haystack relies on an LLM-based metric (Joint Score) to compute the quality of the summary in terms of both the relevance of the candidate bullet points (coverage) and the quality of the generated attribution within the bullet points (citation). The authors note that the metric is recall-based, such that longer summaries are likely to score higher than shorter ones. To account for length bias, the original task instructs models to generate summaries of at most 300 words, which we include in our experiments as well. Specifically, models are instructed in all settings to generate summaries of up to 300 words. We observed that in multi-turn settings, models often *forget* this instruction, leading to non-adherence to the instruction. To avoid penalizing models that correctly remain within the 300-word limit, we truncate summaries that go beyond the limit, removing words in equal proportion from summary bullet points, such that evaluated summaries all respect the 300-word limit. We note that this tendency for LLMs to go beyond is further discussed in Appendix F, where we observe that across tasks, model answer attempts get "bloated" over turns of conversations. In single-turn settings (full, concat), LLMs largely respect the 300-word length limit.

The summary task is a refinement task. Assistant responses at each turn are automatically categorized as answer attempts, and the entire response is considered to be the extracted answer.

### J.7 🅰🈂 TRANSLATION

The Translation instructions were collected from the WMT 2019 task on document-level translation (Scherrer et al., 2019). Specifically, we selected 30 German-English document pairs. Document pairs are aligned at the sentence level (i.e., English and German documents in a pair have the same number of sentences). We truncated the selected pairs to their first ten sentences, and sharded the document instruction such that each shard would introduce exactly two sentences from the document, for a total of five shards. We provided shards in German, and the task consisted of translating into English (i.e., German→English). Hence, Shard 1 introduces the first two German sentences, Shard 2 introduces German sentences 3-4, etc. In the sharded setting, the task requires the LLM to translate the document with all the provided sentences so far. In the full settings, the LLM is provided the entire document (10 sentences) in the first turn. In the concat setting, the LLM is also provided all sentences in the first turn, but separated into shards (two sentences at a time).

In initial experiments, we experimented with other sharding strategies, including breaking shards at a specific number of words (rather than sentence boundary), and increasing the length of documents (from 10 to 20 sentences), without observing significant differences in results. This led us to adopt the setting we describe: sharding every two sentences, and truncating at 10-sentences.

We evaluated performance with the BLEU metric (Papineni et al., 2002), the standard metric for translation tasks, which was used as well in the original WMT 2019 competition.

## K EXAMPLE SIMULATED CONVERSATION

Figure 10 provides an example conversation that was simulated during our experiments in the sharded setting. The simulation was conducted on the Math task, with a 6-shard instruction, and using the Llama3.1-8B-Instruct as the assistant. This conversation illustrates the following properties described in the rest of the paper: (1) the LLM makes assumptions early in the conversation (in Turn 1, describing four pastries that are irrelevant), (2) although it correctly interprets user-provided information, it also unnecessarily updates the information for assumptions it made (Turn 4), (3) this leads to unnecessary complexity, and the model ultimately forgets that the initial instruction was to calculate total calorie count, and returns only half of the calculation (just for Mini Blueberry Muffin). In short, this conversation illustrates the lost in conversation phenomenon: when the user instruction is underspecified (Turns 1-4), the LLM makes assumptions that detract from the conversation and lead to incorrect or incomplete answers.

## L GRADUAL SHARDING IMPLEMENTATION

To evaluate the effect of instruction granularity on performance degradations, we conducted the *gradual sharding experiment*.

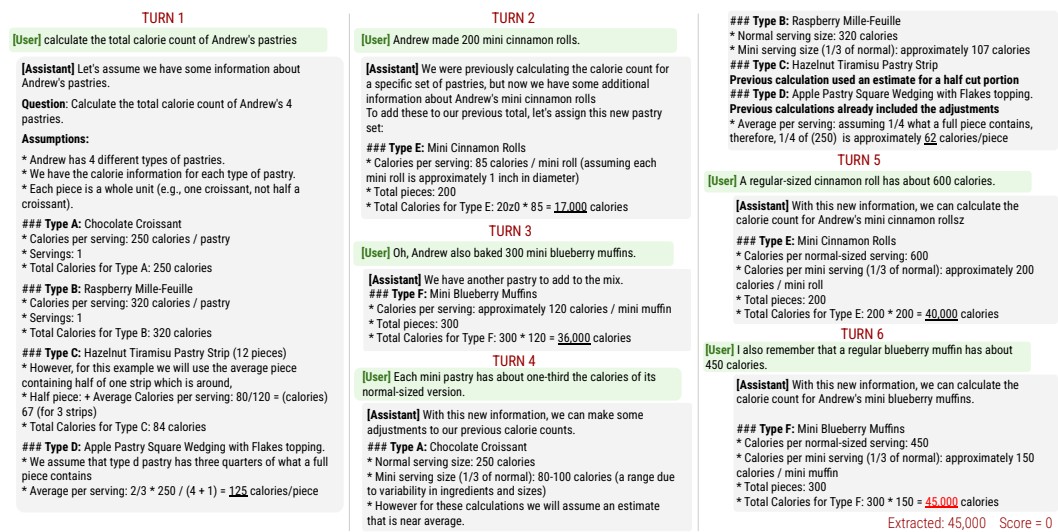

Figure 10: Example simulated multi-turn conversation for the Math task. This conversation simulation was with assistant model Llama3.1-8B-Instruct. The sharded instruction consists of six shards. The correct answer to the instruction is 85,000 calories. Each assistant response was classified as an answer attempt in every turn and was evaluated for early success, but none of the generated answers were correct.

We selected sharded instructions that had exactly eight shards, leading to a total of eight instructions across three tasks (Code, Math, Data-to-Text). We then leveraged an LLM (GPT-4o) to expand each instruction into 7 variants with a differing number of shards. The LLM was instructed to *merge* the original sharded instruction into a smaller sharded instruction with two to seven shards. The instruction authorized minor rephrasing to allow for individual shards to be fluent, but encouraged the LLM to remain as close as possible to the original instruction in wording.

As such, each of the original instructions can be paired to: (1) a concat instruction (one-shard), and (2) 7 sharded instructions, ranging from two to eight shards. Applying this method to the 31 instructions yields a total of 248 instructions, distributed equally in number of shards (from 1 to 8) and on the identical underlying problems.

We ran simulations using the 248 instructions, simulating 10 conversations per instruction and model for two models: GPT-4o and GPT-4o-mini. Findings of the gradual sharding experiment are described in Section 5.3.

## M TEMPERATURE EXPERIMENT IMPLEMENTATION

To evaluate the effect of temperature on aptitude and reliability of LLMs in single- and multi-turn settings, we conducted the following *temperature experiment*.

We selected 10 instructions from each of four tasks: Code, Database, Actions, and Math (for a total of 40). We ran experiments with two models (GPT-4o and GPT-4o-mini). For each instruction and each temperature combination, we conducted simulations for three conversation settings: full, concat, and sharded. For each conversation setting, we varied temperature parameters to three values: 0.0, 0.5, and 1.0. For the full and concat settings, this corresponds to three temperature combinations (as only the assistant temperature can be modified), whereas there are a total of nine combinations for the sharded setting, as both the assistant and user temperatures are varied.

We chose to increase the number of simulations to 20 runs per condition (compared to 10 in the main experiment), as the focus of the experiment is to measure variations in model aptitude and reliability, and adding simulation runs leads to better percentile estimates used in calculating metrics. This added requirement was not computationally expensive, as the temperature experiment involved a limited number of models (2 vs. 15) and instructions (40 vs. 600) in comparison to our main experiment.

Findings of the experiments are described in Section G.2.

## N    Recap & Snowball Experiment Implementation

We leverage SHARDED conversation logs to simulate RECAP setting, since RECAP only differs from SHARDED in terms of an additional recapitulation turn that gathers all the previous user utterances. This implementation also allows us to directly compare the effect of the approach against the SHARDED results. Specifically, for each SHARDED simulation run, we appended the "recap" turn and ran the simulation one more turn. Since it requires stacking the past turns every turn, we simulate the entire conversation from scratch for SNOWBALL simulations. The prompt concatenates the previous turn user utterances as bullet points, followed by the text for the current turn:    We

> **Snowball simulation prompt**
>
> ```
> Just to reiterate:
> - [past utterance 1]
> - [past utterance 2]
>
> Also,
> [current utterance]
> ```

note that what is accumulated for both RECAP and SNOWBALL are verbalized utterances from the user simulator, not the original shards themselves. For both simulation settings, we run $N = 10$ simulations on all of the sharded instructions on four tasks (Code, Database, Math, Actions) and report the mean of averaged performance over the tasks, which is shown in Table 5.

## O    On obtaining deterministic outputs from LLMs

As we demonstrated in our experimental results, setting the temperatures to zero still leads to high unreliability, due to the compounding effect of subtle nondeterminism over tokens and turns.

In theory, greedy decoding (*i.e.*, $T = 0$) will always pick the argmax over the vocabulary distribution. However, it is reported that hardware limitations on floating point operations cause slightly different intermediate values, which results in a ripple effect of larger value changes and therefore different tokens being selected.

Notable model providers acknowledge the non-determinism implicitly or explicitly; Anthropic recommends sampling multiple times to cross-validate output consistency, Google also highlights that their model outputs are *mostly* deterministic, and OpenAI recommends setting the seed parameter to further reduce the non-determinism.

Nevertheless, we caution users that multi-turn conversations can be increasingly unreliable owing to divergent LLM responses.

PROMPTS

## O.1 SHARDING

We show the prompts for the sharding process below, using Math as an example task. Double-bracketed terms are placeholders that get replaced with the actual data. Other tasks share the same outline with different exemplars and rules to enforce stable outputs.

---

**Segmentation**

```
You are a given a fully specified instruction, and your task is to segment the
instruction into a units of information that each reveal a single piece of
information of the instruction.
You must output a list of segments in the following JSON format:
[
    {"segment": "[exact excerpt from the instruction]"},
    {"segment": "[exact excerpt from the instruction]"},
    ...
]

Rules:
- [Non-overlapping] The segments must be non-overlapping and cover the entire
instruction. You can optionally leave some gaps for non-essential portions of the
original instruction (delimiters, headers, etc.)
- [Minimalistic] You should split the information in the segments to as small as
possible. If you have a compound expression (X and Y), you should split it into two
segments. Each segment should represent a unit of information.

Example Query:
What are the names and locations of the stadiums that had concerts that occurred in
both 2014 and 2015?

Output:
{"segments": [
    {"segment": "names and locations"},
    {"segment": "stadiums"},
    {"segment": "concerts"},
    {"segment": "in both 2014"},
    {"segment": "and 2015"}
]}

Now complete the task for the following fully specified instruction:

[[INSTRUCTION]]
```

---

**Rephrasing**

You are given segments of a fully specified instruction, and your task is to: (1) choose one that will be the initial shard of a multi-step query, and then (2) rephrase each segment into a conversational version that are provided to the system in a follow-up turn of the conversation.

Your output should be a JSON object in the following format:
```
{
    "initial_segment": "[exact excerpt from the instruction]",
    "initial_shard": "conversational version of the initial segment",
    "shards": [
    {"segment": "[exact excerpt from the instruction]", "shard": "conversational
    version of the segment taking the rest of the instruction into account"}
    ]
}
```

Example:

Full Query:
What are the names and locations of the stadiums that had concerts that occurred in both 2014 and 2015?

Segments:
```
[
    {"segment": "names and locations"},
    {"segment": "stadiums"},
    {"segment": "concerts"},
    {"segment": "in both 2014"},
    {"segment": "and 2015"}
]
```

Output:
```
{
    "initial_segment": "stadiums",
    "initial_shard": "popular stadiums",
    "shards": [
        {"segment": "concerts", "shard": "the stadiums should have concerts during a
        period"},
        {"segment": "in both 2014", "shard": "the concerts should have occurred in
        2014 in the stadiums"},
        {"segment": "and 2015", "shard": "the concerts should have also occurred in
        2015 in the same stadiums"},
        {"segment": "names and locations", "shard": "for the stadiums, returned both
        the name and location"}
    ]
}
```

Rules:
- [Transform each segment] Make sure each segment is included either as the initial shard or in the rest of the shards. Do not forget any segments.
- [Short initial shard] Make the initial shard short, not a full sentence, similar to how users use a search engine like Google.
- [Order of shards] Order the shards in order of importance, from most to least important to the initial shard. You do not need to keep the order the segments that are provided in.

Now complete the task for the following fully specified instruction and segments:

Fully Specified Instruction:
[[QUESTION]]

Segments:
[[SEGMENTS]]

---

**Verification**

You are given an instruction that fully specifies a problem, and a list of shards.
Your task is to decide whether all the information from the full instruction is
captured by the shards.

If not, you should output the information unit from the instruction that is not
captured by the shards.

Example 1:

Instruction:
What are the names and locations of the stadiums that had concerts that occurred in
both 2014 and 2015?

Shards:
{"initial_segment": "stadiums", "initial_shard": "I'm looking for active stadiums",
"shards": [{"segment": "concerts", "shard": "the stadiums should have concerts
during a period"}, {"segment": "in both 2014 and 2015", "shard": "the concerts
should have occurred in both 2014 and 2015"}, {"segment": "names and locations",
"shard": "for the stadiums, returned both the name and location"}]}

Output:
{"converage": "complete"}

Example 2:
Instruction:
Which Asian countries have a population that is larger than any country in Africa?

Shards:
{"initial_shard": "I'm interested in learning about countries in Asia", "shards":
[{"shard": "consider the population size of these Asian countries"}, {"shard": "the
population should be compared in size"}, {"shard": "specifically, compare to the
population of African countries"}]}

Output:
{"coverage": "incomplete", "missing_segment": "the shards do not specify that the
population of the Asian countries should be *larger* than the population of any
African countries"}

You must output in JSON format as shown in the examples above.
Now complete the task for the following fully specified instruction and shards:

Instruction:
[[QUERY]]

Shards:
[[SHARDS]]

---

## O.2 EXPERIMENTS

The experiments involve several LLM calls with specific prompts to simulate the conversation, which we list below.

---

**Answer Extraction**

```
You are reviewing a multi-turn conversation between a user and an assistant, and are
given the last turn of the conversation.
In the final response from the assistant, a final answer has been provided. Your
goal is to extract verbatim what the answer is:
- If the answer is short (less than 10 words), then you should copy verbatim what
the answer is in the `answer` field.
- If the answer is long, then you should produce the answer with an ellipses, to
indicate the exact start and end of the answer (e.g, ```def funny_function(n): [...]
return funny_output```). You should include *at least* 4 words or one full line for
the start (before the ellipses) and *at least* 4 words or one full line for the end
(after the ellipses), such that the answer can be identified exactly.

Rules:
- [Exact Answer Only] only extract the exact answer, and nothing else (including ```
for code blocks, or intro/outro text).
- [Verbatim Only] Only extract verbatim text, do not modify the text in any way. If
there's a typo, an error, you must absoltutely include it, and not correct it in any
way.
- [Task Specific Answer] [[ANSWER_DESCRIPTION]]
- [String output] the <answer_str> must be a string, not a number and not a
dictionary.

You must output your answer in the following JSON format:
{"answer": "<answer_str>"}

Conversation's last turn:
[[CONVERSATION_SO_FAR]]
```

---

---

**Response strategy categorization**

```
You are reviewing a multi-turn conversation between a user and an assistant, and are
given the last turn of the conversation.

Here is the full specification of the problem the system is attempting to solve:
[[INITIAL_SHARD]]

Specification:
[[SHARDS]]

You must classify the response of the assistant according to the response type:
- `answer_attempt`: The response contains a complete answer attempt to the user's
question (not templated or hypothetical), that can be extracted verbatim. See the
task-specific answer description for more details.
- `clarification`: The response  is short (less than 100 words) and contains a
single question addressed to the user that directly inquires about an aspect of the
user's query. A clarification turn cannot be long (see `discussion`), cannot contain
a vague question (see `discussion`) and cannot contain multiple questions (see
`interrogation`).
- `interrogation`: The response contains multiple questions addressed to the user,
sometimes organized in a list or bullet-points.
- `discussion`: The response discusses the question in detail, without providing a
final answer, asking a specific clarification question, or a refusal to answer. The
response may or may not contain a vague question (e.g., "What else can I help you
with?").
- `hedge`: The response contains multiple answer candidates based on hypotheticals
(ifs) or branching (case 1, case 2) with corresponding descriptions.
- `refuse`: The response contains an explicit or implicit refusal to answer the
user's question without a follow-up question or a request.
- `missing`: The response is empty/blank.

You must output your answer in the following JSON format:
{"response_type":
  "refuse|missing|answer_attempt|hedge\\
  |clarification|interrogation|discussion"}

Rules:
- The assistant giving a hint at how an answer could look like is not a final answer.
You should only select `answer_attempt` if the conversation could end at this stage
with the user having an entirely final answer to the problem they've formulated.
- [Task Specific Answer] [[ANSWER_DESCRIPTION]]

Conversation's last turn:
[[CONVERSATION_SO_FAR]]
```

## User simulator

You are simulating a user of an interactive LLM system (like ChatGPT).
The user is inherently lazy, and answers in short form, providing only minimal
information to the system. You should not be proactive.

Here's the conversation so far:
[[CONVERSATION_SO_FAR]]

Here are the shards that have already been revealed:
[[SHARDS_REVEALED]]

Here are all the shards that have not been revealed yet:
[[SHARDS_NOT_REVEALED]]

You must generate a response to the conversation so far. Here are the rules:
- [Providing a shard] You can reveal the content of a shard to the system in your
response if it will help the system move closer to answering the problem. You should
select the shard to reveal that is most "basic" and currently the most relevant.
- [One Shard at a Time] You should only reveal at most one shard at a time.
- [Reveal Entire Shard] If you reveal a shard, you must make sure to include *all
the information in the shard*. For example, if the shard is "your symptoms are that
you have a headache in the mornings", your response can't just be ``yeah I have
headaches'', you must say ``yup mostly headaches in the mornings``.
- [Irrelevant Clarifications] If the system asks you a question irrelevant to the
shards, asks you a generic question (``Can you give me a hint?``), you should
respond with an answer that does not provide a shard. (``I don't know``, ``Is that
really important?``, etc.) You should not reveal any information beyond what is
available in the shards.
- [No Repeated Shards] You should not reveal the same shard more than once.
Carefully review the already revealed shards, and only reveal a shard if its
`shard_id` is not on the list.
- [Rephrase Shards] If you reveal a shard, you should rephrase it in a
conversational way. Do not copy the shard verbatim.
- [Do Not Ask Questions] Your response should always be declarative sentences, and
not questions.
- [Brevity of Response] You should favor being succint. Your answer can also have
typos, improper grammar, capitalization, etc. You are simulating a real person
talking to an AI, who is in a hurry.
- [Format] Your response should be formatted as a JSON object with the following
keys:
    - `response`: The response to the conversation so far.
    - `shard_id`: The shard you are revealing to the system. The shard_id can be an
    integer, or -1 if you did not reveal any shards.

For example:
{"response": "I don't know", "shard_id": -1}
or:
{"response": "yeah I want it to [...]", "shard_id": 1}

