# OpenReview forum: "LLMs Get Lost In Multi-Turn Conversation"
_ICLR.cc/2026/Conference — ICLR 2026 Oral_

### Official Review · Reviewer_C44a · 2025-10-27

**Soundness:** 2
**Presentation:** 3
**Contribution:** 3
**Rating:** 6
**Confidence:** 4

**Summary:**

This work evaluates large language models under multi-turn and instruction-sharding scenarios (distinct from conventional single-turn evaluations with fully specified requirements). The authors propose a framework that simulates multi-turn instruction sharding by:
(1) Classifying simulation types (single-turn vs. multi-turn, fully-specified vs. sharded, concat, recap/snowball).
(2) Sharding process: extracting multi-round shards from single-round datasets.
(3) Simulation process: using LLMs as user simulators, policy classifiers, and answer extractors.
(4) Evaluation metrics: separating aptitude (upper-bound capability) and reliability (stability) across multiple attempts.

Large-scale experiments across many models show:
(1) All tested open-/closed-source LLMs degrade under multi-turn sharding vs. single-turn complete instructions.
(2) The drop is driven mainly by increased unreliability (90th–10th percentile gap) rather than aptitude loss.
(3) Even coarse granularity causes decline.
(4) A likely cause is that models make early assumptions, generate premature solutions, and then over-rely on them.

**Strengths:**

(1) Multi-turn instruction-sharding evaluation is important and under-explored.
(2) The simulation framework is well-defined; its categorization and evaluation design improve reliability and yield valuable conclusions.
(3) Experiments are broad and thorough, including multi-model, multi-simulation-method, and varying-slice-number analyses.

**Weaknesses:**

(1) Answer-extraction and scoring mechanism is unclear.
- Figure 2 & L152 hint that every “answer attempt” in each turn is extracted and scored, yet Appendix Figure 10 scores only the final answer.
- If each attempt is scored against the full prompt, early correct partial answers may be unfairly penalized; if scored against the current shard, comparability with single-turn baselines is lost.
- Authors should clarify how final scores are computed (average, max, etc.) and what criteria are used for each attempt.

(2) Fairness of early-correct responses.
- If the model answers correctly before receiving all shards, it could be hallucination or redundant information in the original prompt; the asymmetry vs. the single-turn baseline needs justification.

(3) Realism of sharding.
- Atomizing instructions into many non-overlapping single-element shards is artificial; human interactions usually involve 2–3 rounds with decreasing element counts. The current method forces the model to fill context with speculative content, amplifying interference.

(4) Use of reply-content classification.
- The paper mentions passing reply types to the user simulator to adjust the next shard distribution, but the appendix prompt for the user simulator does not include this step, leaving its utility unclear.

**Questions:**

See Weaknesses.

---

> ### Author Response · Authors · 2025-11-20
>
> Thank you for your careful review and feedback!
>
> Q1: Clarification on answer extraction and evaluation. In a conversation containing N shards, the assistant can attempt answering N times – we score the conversation "correct" if any of the candidate answers is marked correct. In other words, we give the assistant as many chances as the conversation keeps on with new shards. If the assistant is able to predict the correct answer before all shards have been revealed (for example by correctly guessing the missing shards), the assistant receives full credit and the conversation ends early. You are correct that Figure 10 did not correctly present the simulation: in fact answer extraction and evaluation is executed at each turn, but in none of the turns did the model produce the correct answer. We will update Figure 10 to present the answer extraction and evaluation result at each turn, to remove this miscomprehension. We hope this clarifies the scoring process, but we are happy to clarify further if not. It should be clear that since the assistant gets multiple attempts in the sharded setting, it technically is favored over single-turn settings, since those simulations have at most a single answer attempt (in the first and only turn).
>
> Q2: Fairness of early-correct responses. The reviewer is absolutely correct that in the sharded experiment, it is possible for the assistant to make an answer attempt that “through luck” is evaluated to be correct. This could be due to many reasons, including: (1) overfitting of the assistant on benchmark problems (since we include instructions from common benchmarks like HumanEval and BFCL in our experiments), (2) the assistant guessing through luck about the missing shards, (3) the evaluation being not thorough and not catching the difference between a partially correct and fully correct solution. These factors give advantage to sharded simulation (compared to single turn settings that have fewer answer attempts by design) and yet we observe large performance degradations in the sharded setting, confirming that simple brute force of answer attempts is not sufficient to score highly in the lost in conversation setting. We will add this point to our discussion, as it relates to the fairness/advantage of the different simulation settings.
>
>
> Q3: Regarding the naturalness of atomic shards. First off, we agree with the reviewer that the simulations we conducted are conceptually simple, and do not reflect all the complexities that can occur in multi-turn exchanges between real users and AI systems. However, we put effort in the sharding process, manually reviewing and editing each shard (see Appendix C), so that the shards represent natural splits of information and do not artificially split information that would more naturally belong together. This involved at times merging shards that had been split, or reorganizing shards as needed. Further the user simulator was implemented with GPT-4o-mini, which was instructed to pick the shard that fit most naturally in conversation (for instance if the assistant asked a clarification question) so that the conversation would be more natural than simply through random slot filling. We will discuss these design decisions, and the need for better a more natural simulation environment in the discussion section. In short, we put in place guardrails to ensure a minimum level of quality, but agree that more work is needed to lead to realistic simulation.
>
> Q4: Clarity on reply types. The reply types are only used to decide whether or not to trigger the answer extraction & evaluation (The forking arrows on the right of "Strategy Classifier" in Figure 2). If the type is "answer attempt," the system will extract and evaluate the candidate answer. Otherwise (i.e., one of the 6 non-answer-attempt types), the conversation keeps on and the next user shard is revealed.
> Our intention was to implement the classifier that follows the previously established response type categories by Herlihy et al., (2024), resulting in 7 reply types.
> We will update the presentation of the simulation walk-through in L147-153 to reduce confusion.
>
> Thank you for pointing this out.

---

> ### Comment · Reviewer_C44a · 2025-11-26
>
> Thanks for the author's response. The current evaluation protocol may overestimate sharding performance, but even so, the sharding performance still appears weak. Therefore, I agree that the overall conclusion remains unaffected. I will maintain my positive rating.

---

### Official Review · Reviewer_ig5S · 2025-10-29

**Soundness:** 4
**Presentation:** 4
**Contribution:** 4
**Rating:** 10
**Confidence:** 5

**Summary:**

Authors present a simulation framework for multi-turn, underspecified conversations called sharded simulation. It converts high-quality single-turn instructions into a sequence of smaller “sharded” instructions that express alltogether the original task. Concretely, a complete query   is rewritten into a multi-turn exchange that begins with an underspecified first turn. Figure 5 illustrates the process. This is a clean, practical way to generate realistic multi-turn queries from single-turn benchmarks.
Using this setup, authors show a consistent “lost in conversation” effect: when models take a wrong turn, they rarely recover. Across tasks and models (from Llama-3.1-8B-Instruct to Gemini 2.5 Pro), average performance drops from ~90% in single-turn to ~65% in multi-turn settings, with declines observed evident in two-turn dialogs. The paper analyzes this failure mode and offers practical recommendations, calling on LLM builders to prioritize multi-turn reliability alongside aptitude.

**Strengths:**

-A very clean experimental protocol: the sharded simulation is carefully constructed and well validated, and the suite of simulation modes (full, sharded, concat, recap, and snowball) is well designed to isolate where and why LLMs get lost in multi-turn conversations. Remarkably also, the authors repeat simulations for each instruction and quantify the resulting variability.

-Large-scale evaluation across six tasks (code, databases, actions, math) with broad model coverage—from open-source to frontier (15 models in total, see tab 1).

-The “lost in conversation” effect is demonstrated clearly.

-Paper very well written and illustrations of tables and figures are extremely good

-The paper culminates with Section 6, which presents concrete recommendations for multiple stakeholders involved in multi-turn conversational agents — including users, LLM developers, and system/chatbot designers.

**Weaknesses:**

-It's a pity the root causes of conversational model failures are buried in the appendix.

-I’m not fully convinced by the loss-of-middle-turns phenomenon (described in Appendix §F.3). This would probably require a deeper and per-model analysis.

**Questions:**

-The example in Fig. 10 is contestable: the expected final answer is 85k calories, yet the assistant never provides a single final total: it reports separate values for the two pastry types. Reiterating the original request (“calculate the total calorie count of Andrew’s pastries”) at the end would likely have produced the correct aggregate. Is this a real failure ? I’m wondering if there would be other types of ‘lost in conversation’ issues.

-Did you verify that, for models with smaller context windows, the full sharded conversation does not exceed their limit? Context capacity varies by model (some are maybe ~64k tokens, some much more), so overlength dialogs could confound the results.

-In the gradual sharding experiments, we observe a performance degradation for N ≥ 2 shards. However, how can we explain the fact that increasing the number of shards (up to 8) does not necessarily lead to higher loss? In particular, the performance for 2 and 8 shards appears comparable in Figure 5c.

---

> ### Author Response · Authors · 2025-11-20
>
> We thank the reviewer for the care in reviewing the work, in particular the time spent reading the Appendix, which is typically optional during reviewing, we appreciate your time!
>
> W1: we agree that the conversational model failures (Appendix F) are indeed important findings of the work. We will dedicate some of the additional content we get (if accepted) to describe each of the phenomena in the main sections of the work.
>
> W2: Regarding loss-in-the-middle, indeed the reviewer is correct that not all models exhibited similar patterns when citing documents from prior turns, with some models having a larger loss of middle turn (typically smaller/older models), and others having more muted effects. For example, some of the top-performing models in long-context settings (such as Claude 3.7 sonnet and Gemini 2.5 pro) did not have a discernible loss-in-the-middle effect. We will add this additional description to the discussion of the loss-in-the-middle result.
>
> Q1: We agree with the reviewer that the example in Figure 10 points to a specific kind of error that occurs due to the assistant solving subtasks and failing to remember to combine it into a single figure, rather than a mathematical/logic error. This is in fact related to our `recap` experiment, in which a final turn is added at the end of the sharded simulation, asking the assistant to attempt one last try, and providing a summary of all the shards in bullet-point form. The findings from the recap experiment Table 5 indicate that though such recap turns help (improving performance from 59.1 to 76.6 for GPT-4o) it still is below single-turn performance (93.0). In other words, recap/consolidation does help the assistant at times to consolidate its correct subtask solutions, but is not sufficient to help explain degradations in performance.
>
> Q2: Regarding the context lengths. All tasks included in our experiments typically require less than 20k tokens in total, with the summarization experiment requiring the most (and all others typically requiring closer to 8k tokens at most). We did include two models (Phi-4, OLMo2) that could not handle 20k tokens, for which we could not conduct the summary experiments (dashes in the corresponding entries in Table 1). For all other (task,model) combinations, the maximum context length is significantly above the requirements for the tasks. We note that the intent of the tasks was not to test model ability in long-context settings (which are known to be limited), but instead to focus on “regular length” but multi-turn settings. We will add this clarification to our experimental setting.
>
> Q3: We agree that the gradual sharding result, and the large drop at N=2 followed by large unreliability from 2-8 shards is surprising. Part of this is due to the fact that the experiment relies on a lower number of instructions, since we only selected original instructions with 8 shards (as we wanted to control for information granularity). These 8-shard instructions are more complex than the average shards in our corpus (with 3-5 shards), and we believe that this filtering led to the selection of more complex instructions. Because of this even the 2-shard version of the shards are complex (since the equivalent of 4 shards is revealed at each turn). A different experiment would involve simply running the analysis with the existing shards in our corpus, and not resharding as we did. This would likely yield more gradual degradation as the reviewer expected, but it would not control task complexity while increasing granularity, as we wanted to demonstrate with the experiment. We will add this discussion on the design decision of the gradual sharding experiment to the paper. In short, understanding the effect of information granularity on an assistant model’s performance is challenging.
>
> Thank you again for your vote of confidence!

---

> > ### Comment · Reviewer_ig5S · 2025-11-21
> > **ack of answer**
> >
> > tks to the authors for their answers to my comments, i read them and have no specific concern about them

---

### Official Review · Reviewer_yCDA · 2025-10-30

**Soundness:** 4
**Presentation:** 4
**Contribution:** 4
**Rating:** 8
**Confidence:** 4

**Summary:**

This paper addresses the critical performance gap between single-turn benchmarks and realistic, multi-turn, underspecified conversations. Using a novel "Sharded Simulation" method, the authors show that LLM performance drops by 39% ("Lost in Conversation"). This drop is attributed not to a loss of core "Aptitude" but to a massive spike in "Unreliability" (performance variance). The analysis identifies root causes like "Answer Bloat" and premature answering. This is a high-impact paper exposing a major flaw in current evaluations and offering a new framework for studying multi-turn reliability.

**Strengths:**

The paper's significance lies in highlighting the critical gap between LLM benchmarks (single-turn) and real-world use (multi-turn).

Its novel methodology, "Sharded Simulation," provides a scalable and clever method to adapt existing benchmarks for multi-turn context evaluation.

The robust experimentation, consisting of large-scale tests (15 LLMs, 6 tasks), provides strong evidence for the findings.

The insightful analysis into "Aptitude" vs. "Unreliability" decomposition is a key insight, pinpointing consistency, not capability, as the main issue, and its root cause analysis (e.g., "Answer Bloat") is actionable.

**Weaknesses:**

The user simulation is a simplification of real, messy human interaction. The reliance on the simulation might weaken the research scope.

Findings are based on analytical tasks; generalizability to creative or open-ended tasks is unclear, and should be considered in future works.

**Questions:**

N.A.

---

> ### Author Response · Authors · 2025-11-20
>
> Thank you for your time reviewing our work and your positive feedback.
>
> We agree with the two limitations the reviewer points to. In the next iteration of the paper, we will add references to ongoing research on user simulation that could improve the realisticness of analyses conducted in our work.
> Relating to the lack of experiments on creative and open-ended tasks, this is a sentiment we share in our Ethics Statement, and we hope the development of improved reward modeling in creative tasks will open the door to simulation experiments in more open-ended domains.
>
> Thank you again for your feedback.

---

### Official Review · Reviewer_RY95 · 2025-11-01

**Soundness:** 3
**Presentation:** 3
**Contribution:** 3
**Rating:** 8
**Confidence:** 3

**Summary:**

“LLMs Get Lost in Multi-Turn Conversation” examines how large language models handle underspecified dialogue by introducing a sharded simulation method that splits single-turn benchmark instructions into smaller “shards” revealed over multiple turns. Evaluating 15 models on six generation tasks across 200,000 simulated conversations, the authors find an average 39% performance drop from single-turn baselines, driven mainly by a large rise in unreliability (+112%) rather than a loss of aptitude (–15%). Control conditions show that rephrasing alone does not explain the effect, and that recap, repetition, or lower temperature settings offer only limited recovery. Qualitative analysis identifies four causes—premature answer attempts, compounding assumptions, over-weighting of first and last turns, and “answer bloat”—together forming the “Lost in Conversation” pattern, where models rarely recover after early mistakes. The authors note that their simulated, turn-by-turn setup idealizes real conversation but likely underestimates the reliability issues seen in practice.

**Strengths:**

### Strength #1: Novel Decomposition of Performance into Aptitude and Reliability

The paper makes an important conceptual contribution by decomposing overall performance degradation into two distinct components: aptitude (best-case capability, measured as 90th percentile performance, A90) and unreliability (variance across runs, measured as the 90-10 interpercentile range, U90/10). This framework reveals that the primary issue in multi-turn settings is not loss of capability but rather dramatic increases in inconsistency.

The empirical findings are striking: while aptitude drops moderately from single-turn to multi-turn settings (average -16%), unreliability more than doubles (+112%). Critically, this pattern holds across all 15 models tested, from small open-weight models to state-of-the-art systems like GPT-4.1 and Gemini 2.5 Pro. This means that even the most capable models become highly unpredictable in multi-turn conversations---they retain the ability to solve tasks (high aptitude) but fail to do so consistently (low reliability).

This decomposition is methodologically enabled by running 10 independent simulations per condition rather than reporting single averages, allowing the authors to measure variance meaningfully. The resulting metrics (P, A90, U90/10) provide a potentially generalizable framework that extends beyond this specific study. Other researchers can apply this lens to different settings (e.g., retrieval-augmented generation, agentic workflows, code generation) to separately diagnose capability versus consistency issues.

The practical implications are significant: the paper argues convincingly that LLM builders should jointly optimize for both aptitude and reliability, rather than treating higher benchmark scores as the sole objective. For end users, the finding explains a common frustration---tasks that should work sometimes fail unpredictably, which may be more problematic than consistent failure. Overall, this decomposition transforms what could have been a purely negative result (performance drops in multi-turn) into a nuanced diagnostic framework with clear actionable insights.

---

### Strength #2: Systematic Behavioral Analysis Identifying Concrete Failure Modes

Beyond demonstrating that performance degrades, the paper provides a thorough behavioral analysis identifying four specific failure modes, each supported by quantitative evidence from the conversation logs. This analysis strengthens the contribution by moving from "models fail" to "models fail because they exhibit these specific behaviors."

**Premature answer attempts** (Section F.1, Table 3): The authors show that timing of first answer attempts strongly predicts success. Across all models, conversations where the first attempt occurs in the earliest 20% of turns achieve only 30.9% average performance, compared to 64.4% when attempts occur in the final 20%. This suggests models jump to solutions before gathering sufficient information, then struggle to revise.

**Answer bloat** (Section F.2, Figure 8): Through measurement of answer lengths across conversation turns, the paper demonstrates that multi-turn answers become progressively longer---20-300% longer than single-turn equivalents by the final attempt. Even correct solutions in multi-turn settings are 14-27% longer than correct single-turn solutions, indicating unnecessary complexity rather than mere verbosity accompanying errors.

**Loss-of-middle-turns** (Section F.3, Figure 9): Using citation patterns in the summarization task, the authors show that models disproportionately reference information from first and last turns (20% citation rate for turn-8 documents) while under-weighting middle turns (8% for turns 2-3). This extends the known "lost in the middle" phenomenon from single-turn long-context to multi-turn conversation.

**Over-verbosity** (Section F.4, Table 4): By binning conversations by response length, the paper establishes that longer responses correlate with worse performance on 5 of 6 tasks. Shortest-response conversations achieve 40.7% average performance versus 35.6% for longest-response conversations, suggesting verbose responses introduce problematic assumptions.

Each failure mode is empirically grounded with specific measurements rather than anecdotal observations, and together they form a coherent narrative: models make premature assumptions when information is incomplete, these assumptions compound into increasingly complex and incorrect solutions, and middle-context information is forgotten while early errors persist. This level of behavioral detail is rare in empirical LLM papers and provides concrete targets for future improvement---for instance, training models to defer solution attempts until sufficient information is gathered, or developing architectures less prone to middle-context loss in conversations.

The analysis is reproducible (the authors plan to release conversation logs) and the failure modes are interpretable, making this a substantive contribution beyond the immediate experimental findings. Future work can test whether interventions targeting these specific behaviors improve multi-turn reliability.

**Weaknesses:**

### Weakness #1: Limited Real-World Validation Restricts Generalizability

The paper’s central claim---that LLMs “get lost in multi-turn conversation”---is supported entirely through synthetic simulations in which single-turn benchmark instructions are artificially fragmented into minimal “shards” (typically 6–8 small facts revealed one per turn). While this setup is useful for controlled stress-testing, the authors do not demonstrate that real human–LLM conversations exhibit similar fragmentation or that comparable degradation occurs in practice. The only empirical link to real data is a citation to Herlihy et al. (2024), which notes that underspecification is common but does not quantify how information is distributed across turns. As a result, it remains unclear whether the reported 39% performance drop reflects natural conversational behavior or an artifact of the experimental design.

The paper’s gradual-sharding experiment (Section 5.3) partly addresses this by showing degradation even in two-turn conversations, but this analysis uses only 31 instructions and 2 models---too narrow to establish generality. Moreover, the authors do not show that their two-turn setup resembles real user interactions any more than their eight-turn cases. Because the title, abstract, and framing imply broad conclusions about multi-turn conversation, readers may overinterpret what is essentially a controlled fragility test. The work would be strengthened by either (1) analyzing real LLM chat logs to quantify natural fragmentation patterns, (2) validating degradation effects through human studies, or (3) reframing claims to emphasize that the results concern performance under idealized underspecification, not natural dialogue.

---

### Weakness #2: Confounded Design Obscures Causal Mechanisms

The experimental setup combines several distinct challenges, making it unclear which factors actually drive the observed degradation. Three design confounds are particularly important:

1. No error correction.
   The user simulator never provides feedback when the model makes an incorrect assumption; it simply continues revealing shards. This means the task tests not only underspecified input handling but also self-correction without external feedback. In real conversations, users---especially experts---often clarify or correct midstream. Without comparing conditions with and without such feedback, it is impossible to disentangle how much of the degradation stems from underspecification itself versus the lack of correction.

2. Minimal conversational context.
   Shards are intentionally atomic (“one fact per turn”), stripping away the pragmatic cues---motivation, examples, and framing---that humans normally provide when clarifying requests. It is therefore unclear whether performance loss arises from the distribution of information across turns or from the absence of contextual scaffolding that would typically accompany clarification.

3. Potentially unnatural communication patterns.
   The one-fact-per-turn design may fall outside the conversational distributions LLMs were trained on. Although the authors argue that their setup is “benign” and likely underestimates real-world difficulty, it might also be adversarial, inducing premature assumptions precisely because it violates ordinary information-packaging conventions.

These confounds complicate causal interpretation. The qualitative analysis identifies four recurring failure modes (premature attempts, compounding assumptions, loss-of-middle-turns, and answer bloat), but the mechanisms remain unclear. For instance, do models guess prematurely because information arrives slowly, or because the conversation lacks corrective and contextual cues? Additional ablations---such as allowing error correction or adding more naturalistic shard phrasing---would clarify which factors truly cause models to “get lost.” As written, the study convincingly shows that LLMs are brittle under a specific, highly controlled multi-turn stress test, but does not yet isolate the underlying causes.

**Questions:**

Q1. How did you decide on shard granularity, and do you think the level of fragmentation reflects natural human–LLM dialogue?

Q2. Did you test whether providing corrective feedback from the user simulator changes the magnitude of performance degradation?

---

> ### Author Response · Authors · 2025-11-20
>
> Thank you for your time reviewing our work and your positive feedback.
>
> Q1: First off, we agree with the reviewer that the simulations we conducted are conceptually simple, and do not reflect all the complexities that can occur in multi-turn exchanges between real users and AI systems. However, we put effort in the sharding process, manually reviewing and editing each shard (see Appendix C), so that the shards represent natural splits of information and do not artificially split information that would more naturally belong together. This involved at times merging shards that had been split, or reorganizing shards as needed. Further the user simulator was implemented with GPT-4o-mini, which was instructed to pick the shard that fit most naturally in conversation (for instance if the assistant asked a clarification question) so that the conversation would be more natural than simply through random slot filling. We will discuss these design decisions, and the need for better user simulation in the discussion section.
>
> Q2: The idea to test corrective feedback is novel and interesting. As mentioned above, the user simulator was implemented in an LLM, and we observed that there are occasions when it generated user utterances in ways that sounded corrective (“No, I want to sort in XYZ way”), however, this was not encouraged in the instruction and was relatively rare. The `recap` experiment (in Appendix G1) is another experiment related to corrective feedback. In this experiment, a final turn of conversation is conducted where the user states that the solutions provided are not correct, restates all the shards in one turn (as a bullet-list) and instructs the assistant to try one last time. The results of the experiment confirm that such corrective feedback did help boost performance a little, but falls short of closing the gap with single-turn performance. We will describe more advanced user simulation requiring more realistic patterns including corrective feedback in the discussion.
>
> Thank you again for your time reviewing our work, we appreciate it.

---

### Meta-Review · Area_Chair_F4Py · 2026-01-08

**Summary:**

The paper investigates why Large Language Models (LLMs) struggle with multi-turn conversations involving underspecified instructions. To evaluate this, the authors introduce Sharded Simulation, a framework that fragments fully-specified instructions from benchmarks (Code, Math, etc.) into smaller "shards" revealed turn-by-turn.

Models show an average 39% performance drop in this simulated multi-turn settings compared to receiving instructions all at once. The authors examined and found that the drop is not caused by a lack of fundamental ability (aptitude) but by a massive increase in inconsistency (unreliability). Models frequently make premature assumptions in early turns and fail to recover once they take a "wrong turn".The authors identify behaviors like answer bloat, i.e., multi-turn answers becoming 20–300% longer than single-turn, and loss of context of middle turns. These finding are fresh and very interesting as acknolwedged by the reviewers.

The review consensus is exceptionally high (10, 8, 8, 6). Reviewers praised the paper for identifying a realistic gap in benchmarking and providing a nuanced diagnostic framework. In rebuttal, the authors clarified scoring mechanisms (favoring models with multiple attempts), verified context window limits, and illustrated the efforts put into manual efforts to ensure shards were natural rather than adversarial. The mete reviewer agrees with the reviewers and recommend acceptance.

**Reviewer Concerns:**

The following concerns are addressed.

- Scoring Fairness: Reviewer C44a questioned if models were unfairly penalized. The authors clarified that the assistant is given as many chances as there are shards, actually favoring the assistant over single-turn baselines; despite this advantage, performance remains poor.

- Context Length: Reviewer ig5S raised concerns about token limits. The authors confirmed the tasks typically require fewer than 20k tokens, well within the context windows of almost all models tested.

- Shard Granularity: Reviewers worried the sharding was unnatural and hindered the model's performance. Authors detailed a semi-automatic process that involves manual inspection and editing to ensure that shards represent natural information splits.

The following concern remains outstanding:

- As noted in the paper’s own limitation section, fully automated simulations cannot capture human nuances such as emotional frustration or misunderstandings of terminology. Reviewers also noted the current evaluation is through synthetic benchmarks. User study or analysis of real-world chat logs of LLMs with real-users are recommended way to alleviate this concern.

**Reviewer Scores:**

All reviewers are very positive about this work, and they will likely remain so after the rebuttal.

---

### Decision · Program_Chairs · 2026-01-26

Accept (Oral)